# Photonic crystal enhanced fluorescence emission and blinking suppression for single quantum dot digital resolution biosensing

Yanyu Xiong[1,2], Qinglan Huang[1,2], Taylor D. Canady[2,3], Priyash Barya[1,2], Shengyan Liu [1,2], Opeyemi H. Arogundade[4], Caitlin M. Race[1,2], Congnyu Che[2,4], Xiaojing Wang[2,3], Lifeng Zhou [2,3], Xing Wang [2,3,4], Manish Kohli[5], Andrew M. Smith [2,4,6,7,8] & Brian T. Cunningham [1,2,3,4,8] ✉

While nanoscale quantum emitters are effective tags for measuring biomolecular interactions, their utilities for applications that demand single-unit observations are limited by the requirements for large numerical aperture (NA) objectives, fluorescence intermittency, and poor photon collection efficiency resulted from omnidirectional emission. Here, we report a nearly 3000-fold signal enhancement achieved through multiplicative effects of enhanced excitation, highly directional extraction, quantum efficiency improvement, and blinking suppression through a photonic crystal (PC) surface. The approach achieves single quantum dot (QD) sensitivity with high signal-to-noise ratio, even when using a low-NA lens and an inexpensive optical setup. The blinking suppression capability of the PC improves the QDs on-time from 15% to 85% ameliorating signal intermittency. We developed an assay for cancer-associated miRNA biomarkers with single-molecule resolution, single-base mutation selectivity, and 10-attomolar detection limit. Additionally, we observed differential surface motion trajectories of QDs when their surface attachment stringency is altered by changing a single base in a cancer-specific miRNA sequence.

Chemical and nanoparticle-based fluorescent reporters are broadly utilized components of life science research and molecular diagnostics. Photon-generating tags enable visualization and quantitation of biological analytes by attaching the reporter to a target molecule, followed by detection with an instrument that excites fluorescence while gathering photon emission into an optical sensor. Colloidal QDs offer a wide range of useful optical properties for digital assays with single-molecule readouts, including large absorption coefficients ($>10^7 \, M^{-1} \, cm^{-1}$), narrow and widely tunable emission bands, high photostability, and high quantum efficiency. Enhancement of fluorescent reporters through locally enhanced electromagnetic field intensities by plasmonic surfaces and nanostructures has proven to be an effective strategy for achieving reduced detection limits in biomolecular assays[1–6], particularly for assays that detect aggregates of many fluorophores, whose emission is combined to yield signals that are detectable above background fluorescence and the shot noise of

[1]Department of Electrical and Computer Engineering, University of Illinois at Urbana–Champaign, Urbana, IL 61801, USA. [2]Holonyak Micro and Nanotechnology Laboratory, University of Illinois at Urbana–Champaign, Urbana, IL 61801, USA. [3]Carl R. Woese Institute for Genomic Biology, University of Illinois at Urbana–Champaign, Urbana, IL 61801, USA. [4]Department of Bioengineering, University of Illinois at Urbana-Champaign, Urbana, IL 61801, USA. [5]Department of Oncology, Huntsman Cancer Institute, Salt Lake City, UT 84112, USA. [6]Carle Illinois College of Medicine, Urbana, IL 61801, USA. [7]Department of Materials Science and Engineering, University of Illinois at Urbana-Champaign, Urbana, IL 61801, USA. [8]Cancer Center at Illinois, Urbana, IL 61801, USA. ✉e-mail: bcunning@illinois.edu

photodetectors[7–10]. Common assay strategies include aggregating many fluorophores together (for example, a microarray spot), or utilizing enzymatic amplification to generate large quantities of fluorescent reporters from a single analyte.

In recent years, a great deal of research has addressed the problem of enhancing the excitation of single fluorescent reporters, efficiently collecting their photon emission, and generating signals that enable them to be observed in the presence of a variety of noise sources. Total internal reflection fluorescence (TIRF) microscopy, for example, can achieve single fluorophore resolution by using expensive high NA oil-immersion objectives and electron-multiplying charge-coupled (EM-CCD) cameras to provide over a 30× increase in signal-to-noise ratio[11, 12]. Plasmonic nanostructures[13–18] have proven successful for localized enhancement of electric field excitation intensity with a fluorescence enhancement factor from ~100 to ~1000, although many plasmonic structures suffer from high non-radiative decay due to intrinsic losses in the metal, quenching, and low directionality of emitted photons[16]. Moreover, the resonance wavelength of such nanostructures is fixed by the size, shape, and material of the nano-resonators. Early approaches for exciting fluorescent reporters with dielectric optical microcavities demonstrate modest emission rate enhancements[19–21]. One limitation of dielectric microcavities is the mismatch between high-$Q$ resonances of the cavity and the spectrally wide emission from inhomogeneously broadened fluorescent emitters at room temperature. Recent reports of electromagnetic field enhancement with plasmonic–dielectric hybrid nano-gap and dielectric nanowire-slabs[22] addressed these issues and show ~ 1000× enhancement, but with a small number of highly localized hot spots that sparsely occupy only a small fraction of the total surface area. Nevertheless, precise alignment between fluorescence emitters and cavity modes is required to achieve such high enhancement factors through sophisticated nanofabrication. To overcome these issues, our previous research used a microscopy-based approach for fluorescence enhancement from a PC surface over extended surface areas. A 60-fold increase in fluorescence intensity has been reported from a Cy-5-conjugated streptavidin layer on a 1-dimensional PC[23]. This enhancement can be improved to 360-fold by coupling the PC leaky mode to an underlying Fabry–Perot-type cavity through a gold mirror reflector[24]. A 108× fluorescence enhancement for a layer of QDs has been reported by using leaky-mode-assisted fluorescence extraction from a 2-dimensional PCs surface[25]. More recently, Yan et al. demonstrated a multiple heterostructure PC with a super-wide stopband to achieve broadband fluorescence enhancement of over 100-fold[26], which required complicated layer-by-layer fabrication of self-assembled 2D colloidal crystal monolayers. Three-dimensional PC structures have also been used to enhance fluorescence. Song et al. spin-coated a Ru dye layer on 3D opal PCs composed of multilayers of PMMA spheres to achieve ~320-fold luminescence enhancement with dual-stopband configurations[27].

While several important classes of low-concentration cancer biomarkers (proteins, circulating tumor DNA, long noncoding RNA, lipids, and metabolites) are currently the topic of intense research activity and clinical validation, here we focus on the detection of circulating microRNA (miRNA, miR). With the prominent rise of liquid biopsy, miRNAs can serve as a promising clinical cancer biomarker, with many studies correlating miRNA concentration to specific health conditions such as a particular cancer type or metastatic state[28–31]. The potential for the sensitive and quantitative determination of the concentration of miRNA biomarkers from human serum on a frequent basis is a step toward early cancer detection, treatment monitoring, prognosis, and prediction of treatment outcomes further emphasizes the need for inexpensive and simple high-performance assays[32]. Our efforts and others have provided evidence that strategically chosen circulating miRNA biomarker concentrations are linked to clinical outcomes. For example, by sequencing RNA contents of plasma exosomes, our team

discovered two miRNAs, miR-375 and miR-1290, that are strongly associated with clinical outcomes in patients with metastatic prostate cancer at the time of developing castration resistance (mCRPC)[33]. The detection of miRNA at a very low concentration and with single-base discrimination without the involvement of RNA sequencing (which requires sophisticated equipment, large sample volumes, and elaborate sample processing) represents an important unmet need in current clinical practice.

Unfortunately, the standard protocol of whole blood RNA isolation and purification followed by target identification by quantitative reverse transcriptase PCR (qRT-PCR) is labor-intensive, requires enzymatic amplification, and can suffer from sequence biases[34, 35]. In practice, qRT-PCR assays require unique primers and amplification methods for short miRNA target sequences, which make them fail for the analysis of small volumes[36], while sequencing-based approaches (RNA-Seq) require elaborate sample processing, expensive equipment, long wait times, and bioinformatics expertise, all of which limit their use. For qRT-PCR[37], high sensitivity is achieved through enzymatic amplification which requires both conversion to DNA (reverse transcription) and enzymatic amplification to completion for accuracy. Digital droplet approaches have significantly improved the quantitative analysis of low-volume biospecimens. However, similar challenges limit the readout of qRT-PCR assays of miRNA in droplet format, compounded by the low throughput of droplet partitioning, equipment cost, small dynamic range, and complex data analysis steps. Electrochemical sensors are capable of ultrasensitive (<1 pM)[38] and amplification-free miR detection with a simple read out[39–41] but have a restricted range of operating temperature[42]. Nonetheless, developing a molecular diagnostic test that is ultrasensitive and highly target-specific is necessary to effectively discriminate nucleic acids of similar sequences at low concentrations. Furthermore, a diagnostic assay that does not require enzymatic amplification is desirable for point-of-care use. The development of rapid and cost-effective diagnostics is essential for disseminating technologies for clinical applications in broad point-of-care settings[43].

In this work, we present a sensing strategy for highly specific detection of a cancer-specific miRNA target by providing digital resolution of individual molecules with an optically enhanced high signal-to-noise ratio. Overall, by quality factor engineering, we achieve a ~3000× enhancement in detected photon intensity from individual QD tags (compared to detection of QDs on a plain, unpatterned glass surface), using experimental characterization supported by electromagnetic simulations to attribute a 23× gain to enhanced excitation, a 39× gain to enhanced extraction (including both photon extraction rate improvement and quantum efficiency change via the Purcell effect) and a 3.5× gain to enhanced collection efficiency. Moreover, the blinking suppression capability of the PC improves the QDs on time from 15% to 85%, thus providing a method to ameliorate signal intermittency issues encountered during ultrasensitive measurements while facilitating fast motion tracking at a single particle level. Herein, by exploiting those synergistic properties, we show the PC-QD system can achieve single QD sensitivity with a high signal-to-noise ratio (~59) using a low NA lens (NA = 0.5, 50×) without TIRF or high gain electron-multiplying camera. Our exploration of physics principles for enhancing the excitation, extraction, emission rate, and collection efficiency from individual QDs is motivated by our desire to develop more sensitive, quantitative, and simple methods for detecting cancer-related biomarkers in low-volume clinical samples. A further aspect of this study is the utilization of single-QD imaging capability for a digital-resolution biomolecular assay that has achieved sensitive and selective detection of a miRNA biomarker which can be adapted to detect other miRNAs as well as DNAs and proteins. In this report, we utilize the PC-QD system to implement a highly specific two-step, room temperature, miRNA assay from a 45 μL sample volume to provide digital resolution of individual target molecules, resulting in ~10 aM detection limit,

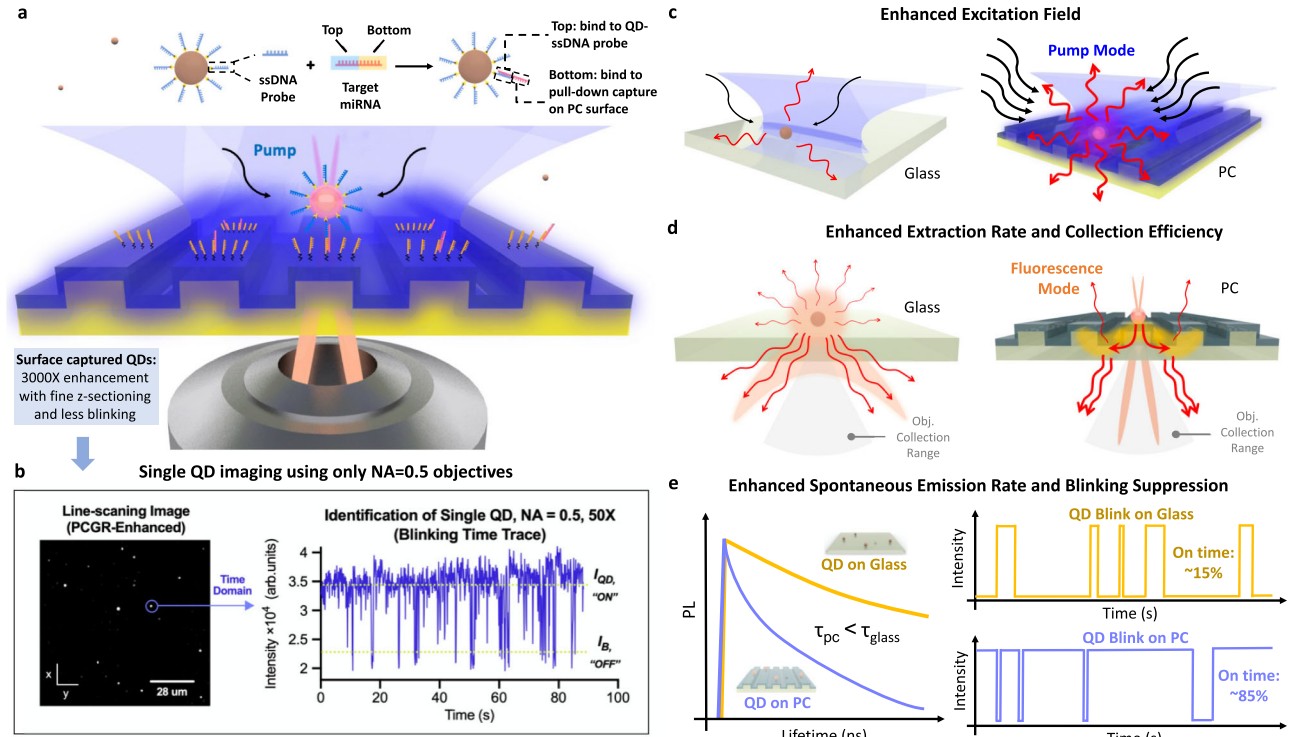

**Fig. 1 | PC enhanced QD emission enables single QD digital counting resolution for single target miRNA detection. a** Schematic of the bridge assay design and PC-QD resonance-enhanced miR digital counting detection approach utilizing 3000× fluorescence enhancement with fine *z*-sectioning and blinking suppression. Components of the PC-QD-enhanced miRNA digital diagnostics are shown in the top part, where ssDNA probe (blue) is functionalized on a QD-streptavidin surface with ~10:1 ratio. Based on NUPACK simulation, the length of the capture strand (yellow) is set to 10 bases (pair with the bottom part of the target) while the probe length is 12 bases (pair with the top part of the target), for a total of 22 bases of complementary pairs for double hybridization with target miRNA 375 (pink). In the bridge-activated assay process, QD-tags will be pulled down to the PC surface when target miRNAs form a surface-bound complex. **b** Surface-captured QDs experience 3000× enhancement compared to the free-floating uncaptured QDs. The strong enhancement factor enables single QD imaging using only NA = 0.5 objective lens.

An example PCEF line-scanning image is shown on the left. Image captured using an EMCCD camera (Hamamatsu) with gain = 1, sensitivity gain = 100 (EM gain 40× out of 1200×), integration time = 600 ms. Laser power = 1 mW. Objective lens: ×50, NA = 0.5. PC pump mode: on-resonance. Time traces of diffraction-limited spot intensities (right) are used to identify single QDs by their distinctive two-level intensity distributions (Objective lens: ×50, NA = 0.5). **c** Illustration of enhanced excitation field. The red arrows denote the QD emissions. **d** Illustration of enhanced extraction rate and collection efficiency. Orange areas show the spatial distribution of QD emissions on glass and PC. The yellow highlights inside the PC indicate the mode profile distribution of its fluorescence mode. QD emitted photons are first coupled to the PC's fluorescence mode which re-radiate with a higher extraction rate to the far-field at an engineered angle that is encompassed by the collection range of objective lens (light gray). **e** Illustration of enhanced spontaneous emission rate and blinking suppression.

single base-pair mismatch selectivity, and high dynamic range (9 orders of magnitude). Interestingly, by utilizing the blinking suppression capability in single particle tracking, our imaging system is capable of recording the dynamic trajectory of single QDs, through which we can discriminate a single base difference in a target miRNA molecule in 10 min without a washing step. Our strategy for simple, sensitive, quantitative, and low-volume miRNA detection is dictated by clinical needs for point-of-care diagnosis. To avoid nonlinearities inherent to multi-step enzymatic amplifications, we use single-endpoint optical detection using QD tags and a PC surface to facilitate high signal-to-noise imaging of individual surface-attached QDs, which enables miRNA to be directly counted.

## Results

### Photonic crystal enhanced quantum dot emission
Advancing beyond earlier reports of quantum dot (QD) enhancement by PC surfaces[1], our present goal is to construct a newly designed PC nanostructured surface that can serve as a general-purpose macroscopic substrate for fluorescence microscopy of QD tags with a simple low-cost fabrication process and greatly improved enhancement factor. As shown in Fig. 1a, b, we designed an instrument capable of enhancing the intensity of QD emission by 3000-fold, opening up opportunities to image single QDs using low NA objectives (NA = 0.5, 50×), without the need for an oil-immersion high NA lens (usually

NA - 1.46) and a sophisticated TIRF instrument. A much greater enhancement is achieved through radiative engineering, where the precise tuning of two quality factors of radiation ($Q_r$) and non-radiation (absorption or inevitable loss caused by fabrication imperfections, $Q_{nr}$) to match each other. $Q$-matching requirements ($Q_r = Q_{nr}$) need to be satisfied for two PC resonance modes (pump-mode and fluorescence-mode), in order to maximize the enhancement factor toward theoretical limits. This 3000× enhancement is also attributed to the integration of multiplicative enhancement factors to provide comprehensive enhancement all the way from the fluorescence generation process (absorption, excited-state lifetime, radiative fluorescence emission) to the collection process (far-field distribution and blinking reduction) after the emission photons are generated.

The interaction of light with fluorescent QD tags can be dramatically modified in the presence of optical resonances[25, 44, 45] through five mechanisms: (i) enhancement of the molecules' absorption (excitation rate) by coupling the excitation pump field into a resonance mode compared with bulk absorption (Fig. 1c), (ii) enhancement of the extraction rate of generated photons into the far-field in the presence of PC compared with placed in the free space (Fig. 1d), (iii) enhancement of spontaneous emission rate and radiative quantum efficiency by modifying the photonic environment of emitters compared with a non-modifying environment[46] (Fig. 1e), (iv) enhancement of collection efficiency by redirecting the emitted light into preferred

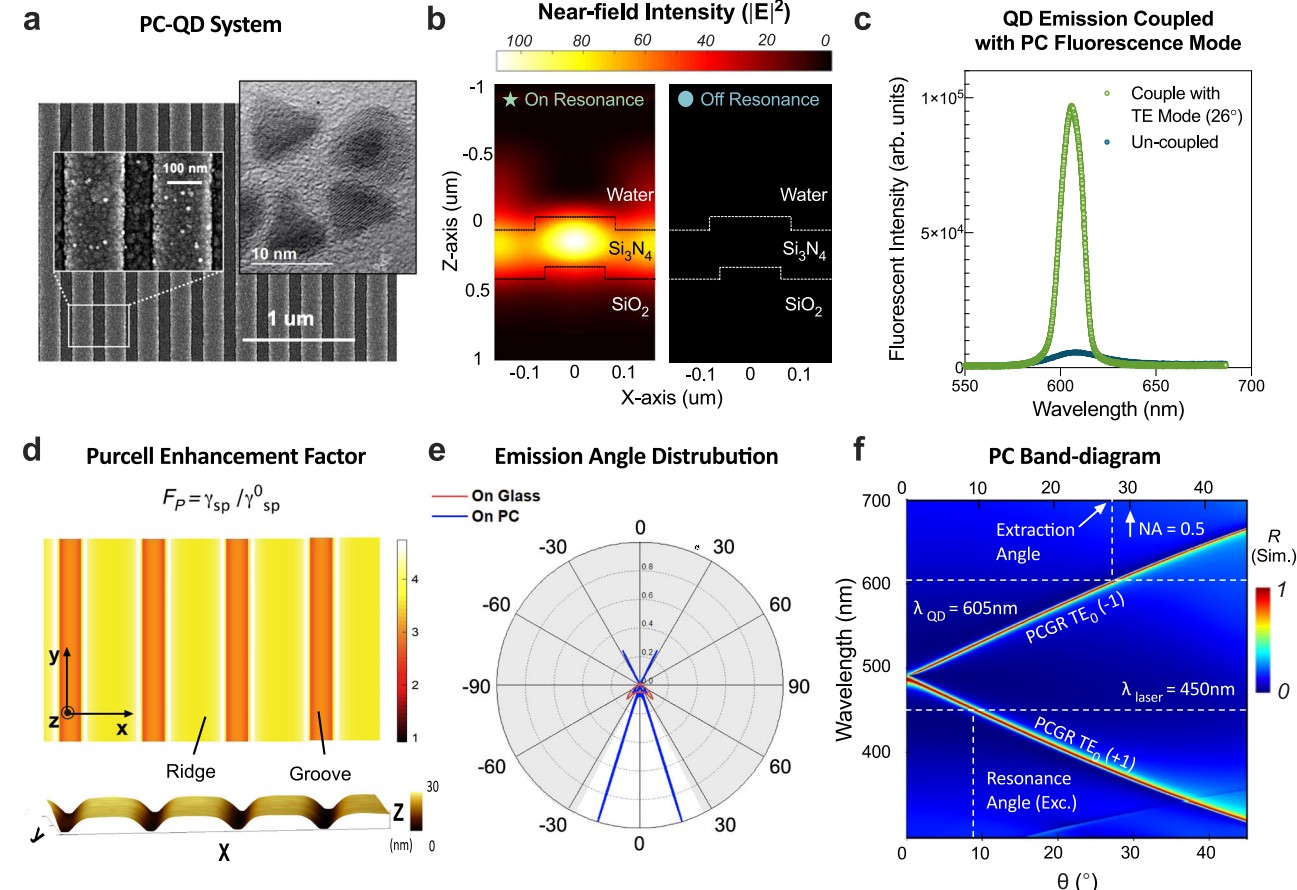

**Fig. 2 | PC-QD system and simulation-based analysis for each enhancement effect. a** SEM image of the PC-QD coupled system at 40k magnification. Scale bar, 1 μm. The left inset shows an enlarged area for detail at 130k magnification with a scale bar of 100 nm. The right inset shows a TEM image of the QD-605 at 600k magnification. Scale bar, 10 nm. More than three experiments were repeated independently with similar results. **b** FDTD simulation of near-field intensity distribution of the PC cross-section, both on-resonance ($\theta = 9.2°$, left) and off-resonance ($\theta = 20°$, right). **c** Angle-resolved emission spectrum measurements for QD-605 when coupled with the enhanced fluorescent mode (green) and solitary on glass (blue), both with collection angle = 26°. **d** Top panel: Numerical simulation for Purcell enhancement factor ($F_p$) 2D mapping as a function of position on the PC surface. A single dipole is used and assumed to be oriented in the $x$-direction. Bottom panel: Topographic imaging for the PC surface using an atomic force microscope (AFM), $x = 1.13$ μm, $y = 0.4$ μm, $z$-height ranges from 0 nm (groove) to 28 nm (ridge). **e** Simulated far-field radiation pattern from the PC surface (blue curve) and glass (red curve). White regions represent angular regions that are collected by NA = 0.5. **f** Simulation of angle-resolved PC reflection spectra (band-structure).

out-coupling directions (Fig. 1d), and (v) blinking suppression (Fig. 1e). First, the PC is designed to support a resonant optical pump-mode at the wavelength of laser illumination, and in turn, to generate an enhanced electromagnetic field for exciting surface-attached QDs. The enhanced excitation mechanism enables substantial absorption of the pump energy for QDs within the vicinity of the PC surface and generates more photons than the same QDs in contact with an unpatterned glass surface. Second, the QD emission light couples with the PC fluorescent mode and radiates into the far-field at a higher extraction rate, so that more photons are captured during a single image integration time. Third, the dielectric PC produces a quantum efficiency enhancement through the Purcell effect, which results in reduced spontaneous emission lifetime and higher quantum efficiency. Fourth, the PC resonant modes provide a photonic dispersion that dictates the highly directional photon angular distribution in free space. The directional emission mechanism ensures greater photon collection efficiency from the PC, through strategic selection of a microscope objective whose NA incorporates the angles for a known QD emission wavelength. Further, we demonstrate that the presence of QDs on the PC results in QD blinking suppression in which the QD remains in its "on" state for a larger fraction of the time. A key element of the current study resides in the combination of several independent multiplicative effects that utilize a PC surface to achieve an

unprecedented 3000× enhancement in QD emission with blinking suppression, which enables observation of individual QDs using a low NA objective while maintaining a high signal-to-noise ratio, a large field-of-view, and improvement of signal intermittency issues.

Excitation enhancement occurs when the PC supports a pump-resonance mode at the wavelength of the excitation laser, which enhances the local electric field (Fig. 1c). The PC-QD system (see Supplementary Fig S1a–e) is comprised of QDs linked with biomolecular bonds to the surface of a dielectric PC slab. Figure 2a shows a representative scanning electron microscope (SEM) image of the PC-QD system. The PC slab supports photonic crystal-guided resonances (PCGRs) in a $Si_3N_4$ thin film ($n = 2.05$) coating on a periodically modulated $SiO_2$ substrate ($n = 1.46$). The surface is immersed in aqueous media, and excited from the backside with a transverse electric- (TE-) polarized laser ($\lambda_{laser} = 450$ nm) at incidence angle $\theta$. When the laser incidence angle meets the phase-matching condition of the PC, it will be coupled as propagating guided modes through first-order Bragg scattering[47]. The excitation light then propagates inside the PC through the resonance-assisted transport pathway and forms 'leaky eigenmodes' that are confined along the PC top surface (extending into the covering aqueous media) for a finite lifetime $\tau$ with tunable Lorentzian line-shape resonance in the frequency domain[19]. With continuous input energy from incident light, the PC serves as a

resonant optical cavity that stores photon energy. As a result, the local excitation field can be orders of magnitude higher than on an ordinary unpatterned surface (Fig. 2b) and can therefore lead to enhanced absorption for surface-attached QDs[48]. At resonance, a strong near-field intensity is built up near the surface and the resonance mode extends across the whole macroscopic surface through the evanescent field tails (Fig. 1c). This delocalized property has opened approaches for the PC to easily interact with added QD tags anywhere across the large PC surface area (in the $x–y$ plane) within close proximity in the $z$-direction (<100 nm, fine $z$-sectioning).

Since the spatial overlap between QDs (average radius ~ 4 nm) and the pump-resonance mode is typically small compared with the wavelength, only a small fraction of the excitation energy is absorbed[45]. Here we define the power absorbed by a single layer of QDs as $P_{Glass}^{abs} = (N_0 \sigma_{QD} d) \times P_{in}$, where $\sigma_{QD}$ is the absorption cross-section of molecules, $N_0$ is the number density of molecules, $d$ is the thickness of the QD single layer, and $P_{in}$ is the excitation pump power. From the temporal coupled-mode theory (TCMT) model[20, 21], we define the excitation enhancement factor to be:

$$\Lambda_{excitation} \equiv \frac{P_{PC}^{abs}}{P_{Glass}^{abs}} = \frac{2\lambda_{laser}}{\pi n d} \frac{\alpha^P (Q^P)^2}{Q_r^P} \quad (1)$$

The ratio term $(Q^P)^2/Q_r^P$ describes how much input excitation energy is stored in the optical resonance through the pump-resonance mode. We applied the energy confinement factor $\alpha^P = \int_{QDlayer} |\mathbf{E}_{\mathbf{k},\omega_k}^P(\mathbf{r})|^2 d\mathbf{r}$ to characterize the proportion of the aforementioned mode-stored energy localized in the QD layer while $\mathbf{E}_{\mathbf{k},\omega_k}^P(\mathbf{r})$ is the normalized electrical field from the pump laser. As the laser is incident through the bottom of the substrate, here the downward radiative quality factor should be considered as $Q_r^P = Q_{r,tot}^P \cdot \frac{S^{P,tot}}{S^{P,down}}$, where $S^{P,down}$ is the downward radiation flux and $S^{P,tot}$ is the total radiation flux.

The second mechanism is extraction rate enhancement due to strong modification of the spectral density of states (SDOS) in the presence of Fano resonances. The spectral and angular emission of QDs can be dramatically altered when coupled to a macroscopic nanostructure resonance compared with that in free space. As shown in Fig. 2c, sharp spectral features in the fluorescence spectra are observed. Similarly, we define the extraction enhancement factor when coupled to QD fluorescing resonance as the ratio between the extraction rate into the far-field in the presence of PC ($\Gamma^{PC} \times \frac{Q^F}{Q_r^F}$) compared with the rate in free space ($\Gamma^{free}$):

$$\Lambda_{extraction}(\mathbf{k},\omega_{\mathbf{k}}) \equiv \frac{\Gamma^{PC} \times \frac{Q^F}{Q_r^F}}{\Gamma^{free}} = \frac{\lambda^F \alpha^F}{n \pi d^F} \frac{(Q^F)^2}{Q_r^F} \cos\theta^F \quad (2)$$

where $Q_r^F$ and $Q^F$ are the radiative and total quality factor of the QD fluorescing channel, representatively. The factor $\Gamma^{PC}$ was defined in a previous study[45] as the rate at which a uniform and isotropic collection of molecules generate photons with crystal momentum $\mathbf{k}$ at the resonant frequency $\omega_{\mathbf{k}}$. We use the total radiative quality factor instead of the downward radiative quality factor because we are going to consider the downward radiation ratio later in the CE calculation.

Moreover, the near-field resonance is directly correlated to its far-field properties. In the far-field, we found the PCGR transport pathway constructively interferes with direct background reflection and results in a reflection peak at the resonant wavelength. Upon inspection of the PC band diagram reflection spectrum (Fig. 2f), we engineer the PC with a moderate quality factor of the pump mode ($Q_r^P = \frac{\lambda_0}{\Delta\lambda} \approx 130.19@450$ nm) when a laser incident at 9.2° matches the QD excitation wavelength. Supplementary Fig. S2 shows the simulated PC reflection spectrum when the PC is on-resonance (light-green star) and off-resonance (dark-green dot) for the pump mode at the QD excitation wavelength. The experimentally measured PC reflection

spectrum is plotted as overlayed gray points with a $Q^P \approx 106.41$. Similarly, the QD emission spectra are compared between PCGR-coupled fluorescent mode (green) and uncoupled emission (blue) in Fig. 1c. With quality factors $Q_r^F \approx 225.08$ (simulation designed) and $Q^F \approx 115.87$ (experimentally measured) at $\lambda_{QD} = 605$ nm. Furthermore, the core–shell structure of a single QD-605 is clearly resolved (Fig. 1a, inset) using a transmission electron microscope (TEM) and the average radius was measured to be approximately 4 nm (Supplementary Fig S7). Assuming a 4 nm-thick QD layer above the PC grating surface, the theoretical excitation and enhancement factor are calculated as $\Lambda_{excitation} \approx 21.84$ and $\Lambda_{extraction} \approx 34.39$ (see Supplementary Part 4).

Before the QD emitted photons couple with the PC extraction mode and leak out to the far-field with a higher out-coupling speed, the process of spontaneous emission is also modified (Fig. 1f). Due to the near-field enhancement and increased density of states, QDs placed on the surface of the PC also experience an enhancement in their spontaneous emission rate. This Purcell effect enhances the QD emission by altering the quantum efficiency (QE) without increasing the non-radiative decay rate due to the lossless nature of dielectric material. The enhanced quantum efficiency for QD-605 on PC (QE$_{PC}$) can be estimated using the following equation:

$$QE_{PC} = \frac{\gamma_{rad,PC}}{\gamma_{rad,PC} + \gamma_{nonrad,PC} + \gamma_{loss,PC}} \quad (3)$$

where $\gamma_{rad,PC}$, $\gamma_{nonrad,PC}$, and $\gamma_{loss,PC}$ are the radiative, nonradiative, and loss rates of the QD on the PC surface. We assume the PC dielectric system does not create additional loss channels ($\gamma_{loss,PC} = 0$) and does not modify the intrinsic non-radiative decay rate ($\gamma_{nonrad,PC} = \gamma_{nonrad,i}$).

The top panel in Fig. 2d shows the 2D mapping of the simulated Purcell enhancement factor ($F_p$) as a function of position on the PC surface. $F_p$ is calculated by the relative ratio of modified spontaneous emission rate modification compared to the original emission rate on glass. Based on the AFM image on the bottom panel, we align the Purcell factor 2D mapping to the topographic structure of the PC grating surface. When the QDs are resonant with the PC fluorescent mode, the ridge regions provide higher Purcell enhancement than the grooves and reach the largest value on the grating edges. This dielectric PC system theoretically can provide 3–5× enhancement to the spontaneous emission rate $\gamma_{SP}^{PC}$ relative to the original emission rate $\gamma_{SP}^0$ on unpatterned glass.

Additionally, the enhanced fluorescent light can couple into PCGR modes through near-field interaction and be radiated to free space for collection by the microscope objective (Fig. 1d). The PC dispersion band diagram (Fig. 2f) dictates how emitted photons will be re-directed toward the microscope objective in preferred directions (enhanced collection efficiency). In our system, the PC has been precisely engineered to offer a dispersion angle for 605 nm emission at 25.8° to ensure that the extracted emission occurs within a limited numerical aperture (30° for NA = 0.5 objective lens).

The far-field radiation pattern and collection efficiency of QD out-coupling are calculated using the finite-difference time-domain (FDTD) method (see Supplementary Parts 6 and 7). The blue and red curves in Fig. 3d present the far-field radiation distribution of QDs on both PC and glass substrates, respectively, in a polar representation. To quantify the enhanced collection efficiency, we define the parameter $CE = \frac{S_{col}}{S_{tot}}$ as the ratio of the collected power that enters the microscope objective ($S_{col}$) to the total power emitted by a single molecule ($S_{tot}$). Simulation results (see Supplementary Part 6) suggest 3.51-fold collection efficiency (CE) enhancement from 5.19% (on glass) to 18.21% (on PC) when collecting emission from below using an NA = 0.5 objective lens. To examine the angular distribution of the emission pattern and its directionality, we imaged single QD emission distribution at the back focal plane (BFP) of the objective lens. Two insets on Fig. 3d show both experimental recorded (blue color

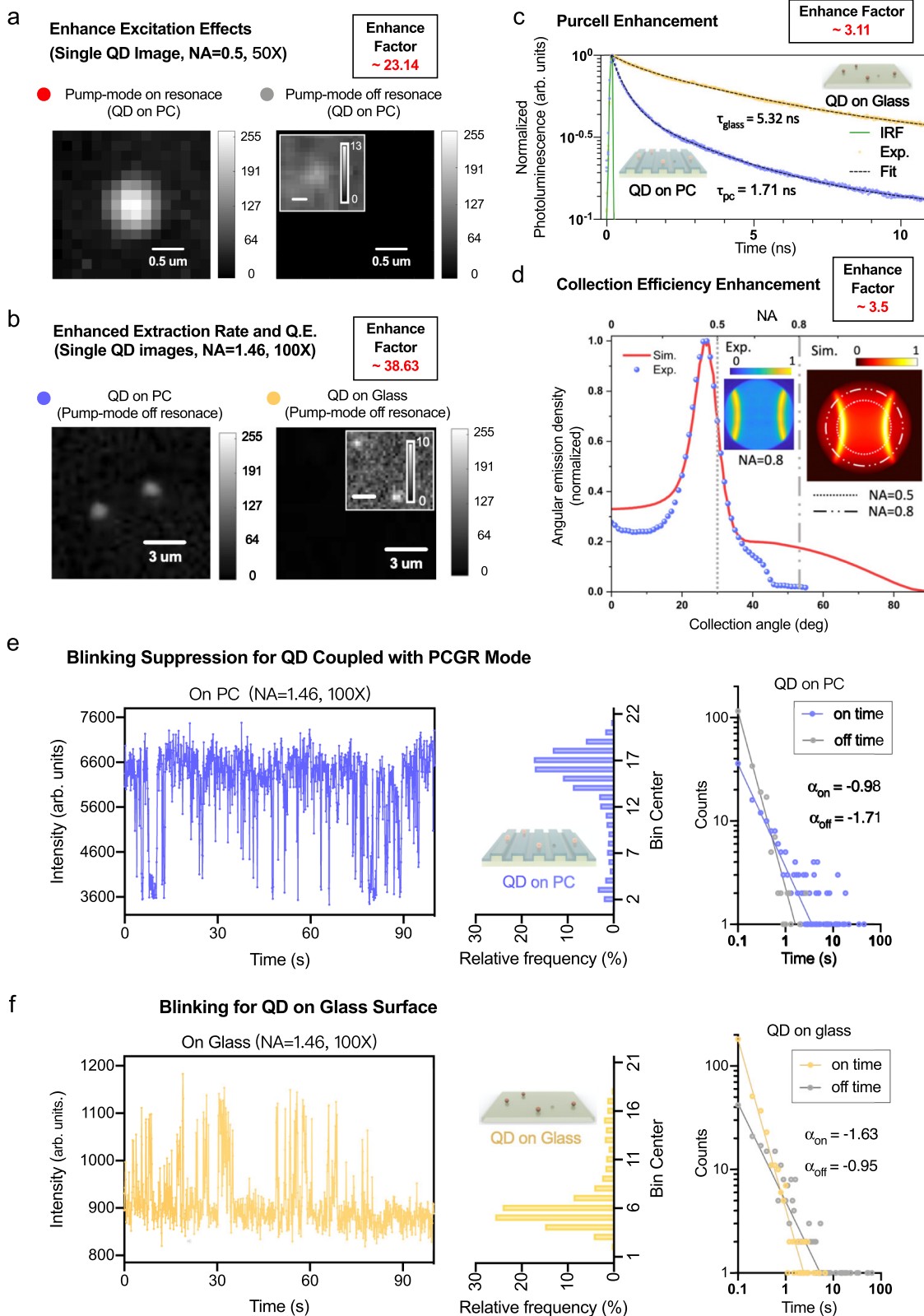

bar) and numerical simulated (red color bar) as double hyperbola-shape bands. The blue dots in the main figure show the experimentally measured normalized power distribution as a function of both collection angle and numerical aperture limits. The red trace in Fig. 3d confirms that the numerical simulation agrees remarkably well with the experimental data. The QD-605 emission (FWHM = 28 nm, see Supplementary Information) is mainly confined in two lobes

centered around 25.8° with a full width at half-maximum of 9.2° angular distribution.

## PC enhanced single QD digital-resolution sensing

Single-molecule digital sensing not only enables a binary decision for each detectable signal but also expands the detection limit of quantitative information by discrete counting for low concentration (sub-

**Fig. 3 | Experimental data for PC-enhanced single-molecule digital sensing.**
**a** Experimental results for enhanced excitation effects. Enlarged single QD image under PCGR-enhanced (left) and unenhanced conditions (right) for pump mode only. The enhancement factor is calculated by $\Lambda_{\text{excitation,exp}} = (I_E - I_0)/I_0$ after background correction, where $I_E$ is the single QD intensity when PC pump mode on-resonance and $I_0$ is the single QD intensity when PC pump mode off-resonance. Since QDs are always on the surface of the PC, the enhancements from extraction, QY, collection efficiency are the same for both cases. **b** Experimental results for enhanced extraction rate and quantum efficiency (QE) effects. Enhanced (left) and unenhanced conditions (right) for both enhanced extraction rate and QE effects. Laser incident angle: 20°. Pump mode resonance off. The collection efficiency enhancement is neglectable when using NA = 1.46, oil immersion objectives. **c** TRPL measurements for ensemble QDs averaged decay time when on glass (yellow) and on the PC (blue). The $\chi^2$ values are <1.2 to ensure the quality of curve fitting. **d** BFP images and angular distribution of the emission. Theoretical (red) and measured (blue) angular emission intensity for a single QD-605 (center at 604.4 nm with FWHM range from 590.4 to 618.4 nm). Insets: Theoretical (red, NA = 1) and experimentally measured (blue, NA = 0.8, in air) back focal plane images of the single QD. The two vertical dashed lines indicate the maximum collection angle of the objectives (NA = 0.5 and NA = 0.8). **e** Blinking suppression for QDs coupled with the PCGR mode. Time trace (left panel, blue), intensity histogram of two distinct states (middle panel, blue), and the on/off time probability distributions follow the power-law for QD on PC (right panel, blue). **f** Blinking for QDs on the glass surface. Time trace (left panel, yellow), intensity histogram of two distinct states (middle panel, yellow), and the on/off time probability distributions follow power-law for QD on PC (right panel, yellow). Integration time: 100 ms, Objective lens: ×100, NA = 1.46, oil-emission. Laser power = 2.1 mW. PC pump mode: off-resonance. All four distributions obey a power law: $P_{(t)} \propto t^{\alpha}$, where $\alpha_{\text{on}} = -0.98$ and $\alpha_{\text{off}} = -1.71$ for QD on PC. $\alpha_{\text{on}} = -1.63$ and $\alpha_{\text{off}} = -0.95$ for QD on glass.

nanomolar) analytes[49]. In order to probe the single-molecule detection capability and the experimental enhancement factors of the PC enhanced fluorescent (PCEF) microscope, a line-scanning experiment was performed while the PC was illuminated at the resonance condition. The laser line intersects a discrete set of individual QDs, which are clearly observable above the background. Specifically, the schematic diagram of the PCEF microscopy setup is illustrated in Supplementary Fig. S3, where a half-wave plate rotates the major polarization of a collimated blue laser diode (TE, 450 nm, 1–5 mW) oriented perpendicular to the PC after passing a linear polarizer. Focused by a cylindrical lens, the incident beam is focused on a line profile (focused in the $zy$-plane while collimated in the $zx$-plane) and aligns to the back focal plane (BFP) of the objective lens (×50, NA = 0.5). The subfigure in the bottom right corner (SI Fig. S3) describes how the incident angle $\theta_{\text{inc}}$ can be precisely tuned by translationally shifting the focus line ($\Delta x$ displacement) on the BFP through Fourier transform. The PC is mounted on a motorized sample stage and moves perpendicularly to the laser line to perform line scanning. The incidence angle selection rule from the phase-matching condition allows direct comparison of the nearfield distribution, excitation intensity, and QD enhanced emission in both the photonic resonator (PC–QD coupled mode) and an unpatterned substrate (solitary QD).

Each diffraction-limited emission location is subsequently examined by analyzing time-domain intensity fluctuations, to demonstrate that the emission originates from a single QD, rather than from QD aggregates. We demonstrate that the PC multi-enhancements are responsible for amplification of the QD emission signal and that the system provides single-QD detection limits through observation of the distinctive binary emission signatures of single-QD blinking. Figure 1b shows the enhanced line-scanning QD imaging across a 120 μm × 120 μm PC surface using the home-built inverted scanning microscope[30] with precisely tuned incident angle (-0.025° accuracy) to the support on-resonance PCGR mode. The incident beam is focused into a line (1.5 μm × 0.5 mm) oriented along the $x$-axis on the PC surface (perpendicular to the grating) and scanning along the $y$-axis. An automatic angle scanning test is performed to ascertain the exact resonance angle in the test region and to thus ensure the angle-sensitive enhanced excitation.

The excitation amplification factor of single QD imaging was compared between both PCGR-enhanced (pump mode on-resonance, Fig. 3a) and unenhanced cases (pump mode off-resonance, Fig. 2b, $\theta = 20°$). The experimental resolved average enhancement factor of $\Lambda_{\text{excitation,exp}} \approx 23.14$ (sample size of 100 individual QDs) matches the simulated result in the previous section. The inset of Fig. 3a shows the linear contrast-enhanced image for the same pump mode unenhanced picture. Without the excitation enhancement from the PC, we are unable to clearly resolve any QD's signal over the relative noisy background using the same excitation power (1 mW), objective lens (NA = 0.5), and CCD integration time (0.6 s) utilized in our system. These results go beyond previous reports[50,51], showing that single QDs can be observed using a low NA objective lens (only 0.5, compared to 1.4–1.46 in prior reports) and normal inverted microscopy with multiple enhancements from the PC surface. Single-molecule binding interactions have been further characterized by utilizing the QD blinking phenomenon[50,52–54]. The time-domain intensity trace (Fig. 1b) was recorded for a series of images gathered while keeping the laser line in a fixed position with an integration time of 100 ms. The pixel intensity is calculated by the average value (16-bit raw data) of a single diffraction-limited bright spot. Two distinct intensity states ($I_{\text{QD}}$ and $I_B$) can be clearly distinguished due to on-and-off intermittency of emission and correspond to the intrinsic QD intensity and overall background. We attribute the significant gap between QD signal and background to the extraordinary signal-to-noise ratio improvement (SNR = $\frac{|I_{\text{QD}} - I_B|}{\sigma_{\text{noise}}}$, increased from <1 to 59.31) offered by PC surface enhancement.

Since the enhancements to extraction rate and quantum efficiency are always present for a QD located on the PC surface, we probe the experimental enhancement factors ($\Lambda_{\text{extraction,exp}}$ and $\eta_{\text{QE}} = \frac{\text{QE}_{\text{pc}}}{\text{QE}_0}$) by comparing single-QD imaging when the QD is on the PC (while pump mode is off-resonance) and on a glass surface. Larger NA (1.46, oil immersion, ×100, Zeiss α Plan-APO-CHROMAT) and higher laser input power (2.1 mW) are used in order to observe single QDs on the glass surface. Figure 3b shows the comparison between the single QD emission difference gained from enhanced extraction and enhanced QY when single QDs on PC (left) and on glass (right). Here, the laser was incident at an angle of 20° (off-resonance for pump mode) to eliminate the enhanced excitation effects from the PC. The enhancement of collection efficiency is neglectable when using the high NA objective. Thus, the factor of enhanced extraction and QY is measured by $\Lambda_{\text{extraction,exp}} \cdot \eta_{\text{QE}} = \frac{I_{\text{QD}}^{\text{PC}} - I_{\text{QD}}^{\text{Glass}}}{I_{\text{QD}}^{\text{Glass}}}$ after background correction. Average extraction enhancement factor of ~38.63 has been calculated based on a sample size of 100 individual QDs, which is comparable with the aforementioned simulation data.

## Purcell enhancement and blinking suppression

Similar to the numerical simulation, our time-resolved photoluminescence (TRPL) measurement indicates an experimental Purcell factor of ~3.11 for ensemble QDs on the PC surface compared to glass. Shown as Fig. 3c, the averaged decay time of $\tau_{\text{glass}} = 5.32$ ns (±0.24 ns) while the $\tau_{\text{pc}} = 1.71$ ns (±0.31 ns). The calculated emission rate depends on the lateral position of the QD on the PC surface and remains constant for a broad wavelength range (500–700 nm) of single fluorescent emitters (see Supplementary Fig. S6). The intrinsic quantum efficiency ($\text{QE}_i = 40.84\%$) for QD-605 on glass slides obtained in (Supplementary Part 9 and Fig. S5) was used to determine the ratio of the intrinsic non-radiative and radiative decay rate ($\gamma_{\text{non-rad,i}}/\gamma_{\text{rad,i}} = 0.4084$). Using Eq.

(3) and the experimental resolved average Purcell enhancement factor ($\gamma_{rad,PC}/\gamma_{rad,i} = 3.11$), we can estimate the $QE_{PC}$ to be 68.22% and the QE enhancement factor ($\eta_{QE} = QE_{PC}/QE_i$) to be 1.67.

The local density of available photon states is increased in the QD–PC coupled system compared to a solitary QD. This accelerates the radiative decay at which an excited state QD will transition to the ground state with a reduced lifetime (faster decay rate) and emits photons spontaneously. The higher rate of this radiative decay process competes favorably against the other nonradiative decay pathways by which an electron is lost through Auger recombination and/or surface trapping which results in off-states[55, 56], thereby suppressing blinking. Figure 2h shows the blinking suppression for a QD coupled with the PCGR mode. The threshold is defined as twice the mean background level. We calculate the fraction of on-times is 87.21% on PC and 14.49% on glass. Similar values were obtained when defining the threshold as the minimum between the two on- and off-states distinct peaks. The probabilities of the on and off periods for a QD on both PC (blue) and glass (yellow) were fit to a power-law distribution of the form, as shown in Fig. 3f, the power parameter for the "on" time, $\alpha_{on} = -1.63$, for QD on the glass case, while the "off" parameter, $\alpha_{off} = -0.95$. For the QD on PC (Fig. 3e), the on and off probability parameters were $\alpha_{on} = -0.98$, $\alpha_{off} = -1.71$, showing an increased probability of on events and decreased probability of off events.

## miRNA bridge assay design and digital counting detection approach

Having demonstrated the ability to gather high signal-to-noise images of single QDs immobilized on a PC surface, we next sought to utilize QDs as biomolecular tags for an enzyme-free miRNA assay, in which each target miRNA molecule in the test sample can result in the attachment of one QD tag to the PC surface. We refer to this type of detection as a digital resolution because each QD may be counted to yield a direct quantitative measure that does not require enzymatic amplification of target molecules, in which low concentration target molecules are expected to result in tags that are distributed at low density across the detection surface. Further, we desire a molecular biology approach that is highly selective for the target miRNA sequence, which will result in the non-capture of QD tags for non-target sequences.

Following recent work associating miR-375 levels in human serum with prostate cancer metastasis, aggressiveness, and prognosticate survival[57–60], we selected this biomarker as the initial target for characterizing the capabilities of our detection platform. In Fig. 1a, the PC was utilized as the biosensor substrate for a bridge-activated assay in which opposite sides of the miRNA target molecule bind with complementary single strand nucleic acid sequences immobilized on the QD and on the PC. Thus, two high specificity nucleic acid binding interactions must occur for a QD to make a strong surface attachment to the PC, and the miRNA target molecule serves as a biomolecular bridge between the QD and the PC (Fig. 4a).

The PC was initially prepared by evaporative deposition of a uniform epoxy-silane layer. Before silanization, the PC top surface was treated with oxygen surface plasma for 10 min in order to activate the surface by creating a density of –OH groups as well as removing organic contamination. The miR-specific DNA probes and target oligos shown were designed using NUPACK[61] (see Supplementary Part 15). The pull-down capture nucleic acid probe sequence (yellow) is a single-stranded DNA (ssDNA) with an $NH_2-$ (amino radical group) modification at the 5′ end for covalently attaching to the silanization layer for PC surface functionalization. ssDNA probes (blue), are conjugated with QD-tags (68 nM) through a biotin–streptavidin interaction and purified by repeated filtration. Moreover, we design QD-tagged ssDNA probes and capture sequences with 5-base oligo(T) linkers as spacers to prevent steric effects between the QD and PC surfaces.

The miR-375 detection target (pink) includes 22 nucleobases, where the 5′ end (10 bases) is complementary to the capture sequence on the PC surface (yellow), and the 3′ end (12 bases) is complementary to the QD-ssDNA probe sequence (blue). The complementary nature of this base-pair structure enables the target miRNA to act like a bridge that stabilizes the connection between capture and QD-ssDNA probe by forming a DNA–RNA duplex. Without the bridge—the target miRNA —the fluorescent tags (QD-ssDNA probe) will not be pulled down to the PC surface, and thus will not experience the aforementioned 3000× enhancement, even if they are illuminated by the excitation laser (Fig. 4b). Following recent guidelines in specific and robust hybridization probe design, we optimized the reaction Gibbs energy between the DNA probe and the target to be approximately zero ($\Delta G' \sim 0$)[62]. At $\Delta G' \sim 0$, the average energetic penalty of a single mismatch is larger ($\Delta\Delta G$) than the free-energy gain of the perfect match, thereby limiting the off-target binding with near-optimal specificity and providing 99.2% hybridization yields (at 23 °C, $Na^+$: 1 mM). NUPACK was used for simulating the equilibrium base-pairing properties in terms of nucleic acid secondary structure and free energy. The number of nucleic acids of both the capture and ssDNA probe was carefully chosen to minimize undesired secondary structure formation. For example, if the number of bases of the capture sequence increases from 10 to 12, the capture oligos form a hairpin loop structure and have no ability to pair with the bottom part of the target miRNA.

Since the PC exclusively enhances the QD-tagged probes with close proximity to the surface through evanescent field enhancement and enhanced guided extraction, PCGR enhanced microscopy provides TIRF-like z-sectioning that enables only QD tags within ~60 nm of the surface to be counted. Consequently, only the pulled-down QDs will be enhanced and report a positive signal for individual binding events; while the unbound free QDs will not be counted since the unenhanced single is comparable to background noise. Moreover, both the PC surface and QD surface are anionic, bound by ssDNA sequences, and tend to electrostatically repel one another to prevent nonspecific adhesion of the QDs to the PC surface. The aforementioned 2-step assay was performed by sequentially incubating a defined concentration of miR-375 (4 h), followed by adding a constant amount of ssDNA-QD probes (2 h) in PC-adhered polydimethylsiloxane well (~45 μL per well). Each incubation step is accompanied by gentle washing steps to remove the free targets and unbound QD-tags.

## Direct counting of miRNA and dose–response curve

Conventional surface capture assays can measure fluorescently labeled analytes across a 1000-fold concentration range and at the sub-nanomolar level, but many biological molecules exhibit more than 1,000,000-fold variations in abundance down to the sub-femtomolar level. Digital PCR amplification-based assays can offer absolute quantification for nucleic acid with an improved dynamic range up to five orders of magnitude[63–65] using precise thermal cycle control. Without enzymatic amplification, our previous research[66] combined single-molecule counting and intensity calibration showing the dynamic range of fluorescence assays can be expanded to a similar range ($10^5$-fold) and reach ~10 fM detection limit using a TIRF microscope with a NA = 1.46, ×100 lens and high-gain EMCCD. Ultimately, we demonstrate the multi-enhancement from the PC–QD assay can provide enzyme-free digital resolution biosensing for direct counting of single miRNA with inexpensive optics (NA = 0.5 objective, low power laser diode, and low EM gain CCD). The direct counting capability yields linear dose–response characteristics when plotted on a log–log scale across a target concentration range of 10 aM–1 nM with 10,000× detection limit improvement compared to analog ensemble intensity measurement.

Circulating miRNAs are highly stable in human serum[67], and the direct detection of miR-375 in serum was recently shown possible[68]. To

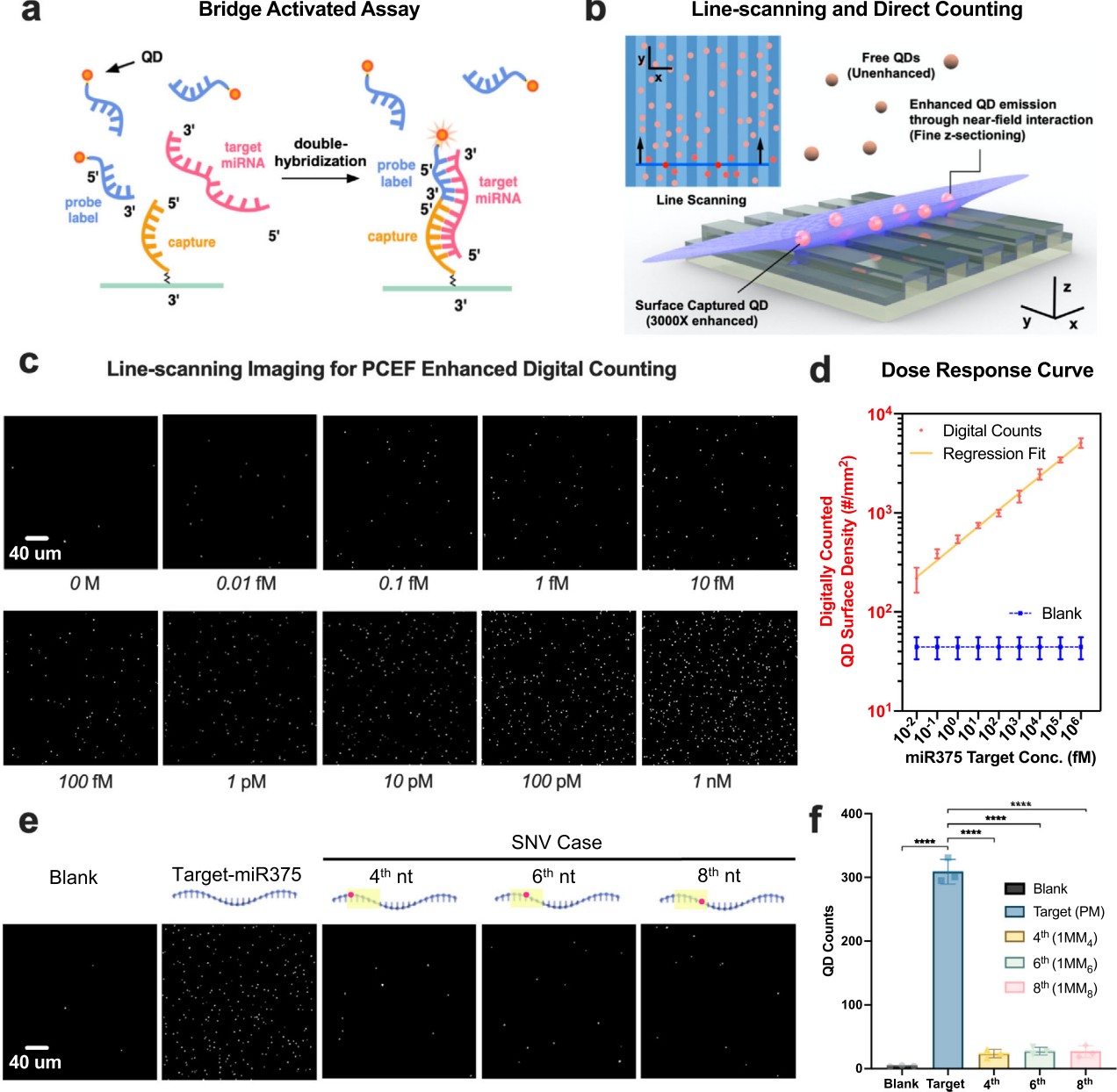

**Fig. 4 | Counting result of miR375 bridge assay in human serum. a** In the bridge-activated assay process, QD-tags will be pulled down to the PC surface when target miRNAs bridge the formation of a surface-bound complex. **b** Illustration of the line-scanning process that counts the number of PC-attached QDs. **c** Line-scanning Imaging: for PCEF enhanced digital counting in human serum. Target concentration: 10 aM to 1 nM. FOV: 300 μm × 300 μm. Scale bar: 40 μm. Data are averages from more than 9 FOVs, and error bars indicate the standard deviation between three independent replicas. Statistical significance was tested using one-way ANOVA between the negative control group and all testing groups with $P < 0.0001$; Imaging conditions for the assay on the PC: laser power = 1 mW, EM-gain = 40×; Higher laser power (5 mW) and EM-gain (1200×) are needed in order to detect the intensity change at 1–10 pM on glass while avoiding signal saturation at 1 nM; Same integration time (600 ms) and objectives (×50, NA = 0.5) were applied in both surface assays. **d** Dose–response curve for various concentrations across a $10^9$-fold concentration range after a 2-h incubation for digital counting results. The error bars represent the mean and the standard deviation of $n = 3$ independent assays. **e** Single-base mismatch discrimination test. Line-scanning image panel demonstrates the digital resolution of captured QDs for the target miR375 perfect match (PM) group, versus three different SNVs at 100 pM. **f** Quantification of the perfect match target sequence and the SNV cases at 100 pM concentration. Statistical significance was tested using independent $t$ tests (two-sided); ****$P < 0.0001$. The error bars represent the mean and the standard deviation of $n = 3$ independent assays.

demonstrate the feasibility of our QD-based digital sensing in an unprocessed native sample matrix, we spiked miR-375 targets in human serum (Sigma-Aldrich, H3667) at 12 final concentrations ranging from 100 zM to 10 nM, and then carried out the sensing assays without performing RNA extraction or purification. Figure 4c, d shows the PC enhanced digital resolution counting results in comparison with conventional analog intensity measurements from a glass surface-based assay (SI Fig. S14). Line-scanning results are selected

representations of the aforementioned assay for specific concentrations of target miRNA in buffered solution. The line-scanning microscope was programmed to acquire the QD emission light from each pixel within the field-of-view (FOV) for QD-tag counting or mean intensity recording. For each testing well, we scanned the total surface with the FOV = 1 mm × 1 mm and divided it into 9–11 small regions (sub-FOV = 300 μm × 300 μm). In order to obtain representative values, the reported count for each group is the average of all 9–11 sub-FOVs.

Several imaging analysis algorithms have been employed, including Bradley local thresholding[69], adaptive contrast enhancement[70], and size-dependent particle identification. Figure 4d shows the captured QD count after 2 h of incubation of the test sample with the PC (at binding equilibrium) as a function of serially diluted miR concentration, with 10 aM and 1 nM representing the lowest (excluding no miR) and highest concentrations measured, respectively. The error bars shown in Fig. 4d represent the standard deviation obtained from three biological replicates within an individual liquid compartment. Three independent double-blind experiments have been carried out during target concentration labeling, assay incubation, and particle counting to avoid cherry-picking and minimize other anthropic factors. As expected, negligible QD background binding was observed with 0 M miRNA-375 (reference group in Fig. 4c) over the entire imaging time duration. With a significant counting difference between 3-times-standard deviation at 0 M, a limit-of-detection (LOD) of 10 aM is demonstrated. Further reduction in target concentration (1 aM and 100 zM) leads to counting results at the same level of the background signal as the availability of the surface captured target is now limited by the sample volume (see SI part 15). Without the PC-assisted single target counting, the conventional analog intensity measurements require higher laser incident power (5 mW) and the highest EM-gain setting (1200×) from the CCD camera to reach an LOD of 1–10 pM (SI Fig. S16).

### Demonstration of high selectivity for single-base variation using both static and dynamic detection

In order to demonstrate the assay's selectivity, we simulated the single-base mismatch variation (single nucleotide variants, SNVs) for all 22 positions along the miR-375 target using NUPACK. Results (SI, Fig. S17) show that our bridge assay is selective against the single-base variation on position index 4-8 (counted from the 5′ end), where this specific mutation sits has been reported to have clinical significance[71]. Figure 4e, f illustrate a dramatic decrease in particle counting, resulting from SNVs at index #4, #6, and #8, with a range of 90–92% signal reduction (difference in counting results: $\triangle N = \frac{N_{PM} - N_{MM}}{N_{PM}} = 90-92\%$, with two-tailed $P$-values of <0.0001) compared with the perfect-match group (PM) at 2 h.

In addition to probing the end-point value when the bridge attachment reaction reaches equilibrium, we also compared the transient interaction between the QD and the PC surface by analyzing the dynamic characteristics of the motion trajectories of QDs that are sampling the PC surface during the process of binding. In order to increase the system's overall free energy and extend the interaction duration, we reduced the ratio of ssDNA probe per QD from 10:1 to 5:1. We initiate the recording of fluorescence image sequences immediately after adding the ssDNA-functionalized QD probes. Starting with a blank surface, we observed QD emission in our imaging field of view after -10 min of incubation with the test sample. A single-particle tracking algorithm was implemented using MATLAB u-track software[72], and each particle trajectory was analyzed for diffusion. We observed distinct trajectories of surface-engaged QDs and selected two representative images, as shown in SI Fig. S15, to demonstrate their unique fingerprint caused by binding discrimination. We hypothesize that manipulation of the dissociation constant of the miRNA–ssDNA interaction for the bridge assay will result in variability in the surface-localized diffusion path. Following the standard method of single-nanoparticle tracking, we further analyzed the trajectories by calculating the time-dependent mean-square displacement (MSD). The resulting trajectories are usually analyzed for their type of motional behavior as the motion provides information on the interactions between the particle (QD-probe) and its surroundings (different target substrates). Five linear MSD plots (green) in SI Fig. S15 indicate normal diffusion when the target substrate has a single-base mismatch and can be described by $r^2 \propto D\tau$ ($r^2$: diffusion displacement, $D$: the diffusion coefficient, and $\tau$: time interval). Conversely, the MSD of the perfect match group (blue) asymptotically approaches a maximum value for larger $\tau$, indicating that the QD-probe is undergoing confined or corralled diffusion with a $r^2 \propto 1 - e^{-D\tau}$ relation.

## Discussion

By utilizing PC enhancements in both near-field QD excitation and far-field dispersion-guided outcoupling, we developed a simple photonic biosensor platform that provides amplification of QD emission of up to 3000-fold. We were able to resolve single QDs even with a low numerical aperture of 0.5, enabling an inexpensive optical setup to provide digital resolution of individual target molecules for highly specific, broad dynamic, rapid, and clinically relevant attomolar miRNA detection. Our PC–QD imaging platform does not require a high gain of an EMCCD camera (×40, only 1/30 of the maximum EM gain on the camera) and further reduces the cost and complexity of the instrument.

The bridge assay was conducted at room temperature without enzymatic target amplicon, representing a simple workflow compared to qRT-PCR. The small mass of QD tags serves to reduce gravity-driven precipitation, resulting in low background counts (from non-specific binding or adsorption)[73]. Pre-incubation of target-capture helped to ensure each surface attached QDs contains only one single target molecule, and hence provide a linear dose–response curve correlated to the extensive range of analyte concentration (0.01 fM–1 nM) and thus facilitate quantitation without the need for post-detection correction by Poisson statistics analysis[74]. We observe a linear dose–response plot when our data is plotted on a log–log scale (Fig. 4b). The observed behavior is the result of several factors that include: (a) the limited diffusion[75] of the miRNA to the biosensor surface, where they can be captured, especially at lower concentrations and (b) the steric hindrance and surface saturation, where the fraction of available binding sites on the PC surface is reduced at high concentrations, making it more difficult for newly arriving miRNA to bind[76,77]. Importantly, we note that the dose–response plot (Fig. 4b) demonstrates no significant overlap between adjacent concentrations based on the standard deviation values for three independent measurements at each concentration. Thus, the digital resolution detection approach shown in our work does not require Poisson correction, as needed for approaches such as ddPCR and Quanterix Simoa™, where multiple target molecules are confined in the same nanodroplet volume are amplified together while still yielding only one positive event.

Single-nucleotide variants (SNVs) can be discriminated in static counting after 2 h with high spatial detected signal reduction. As a further step, we probed the DNA-hybridization kinetic and transient interaction by using dynamic trajectory analysis of single QDs and demonstrated the ability to discriminate SNVs in 10 min without washing, which may offer a further route towards differentiating whether individual tags are bound specifically or nonspecifically. Interestingly, with the combined enhancement effects, we can observe QD saturation and multiexciton generation when the input laser power is higher than 3 mW under PC on-resonance conditions at room temperature (Supplementary Fig. S7).

Our PC-enhanced QD emission platform is not strictly limited to a single wavelength. By engineering the spectral overlap and dispersion properties of the various leaky modes supported by the PCs, one can extend the enhancement effect to a wide range of fluorescent species that utilize the same excitation wavelength (for example, QD-525, QD-565, QD-585, QD-625, QD655, QD685). Although not the focus of this report, we anticipate that the approach described here can be easily multiplexed through future use of QD tags with distinct emission spectra, in which each QD color can represent an assay for a different miRNA target sequence in the same test sample. In future work, we will combine multiple target miRNA sequences with distinct co-

immobilized capture DNA sequences to enable multiplexing within a single test region.

## Methods

### Photonic crystal fabrication (PC)

The PC fabrication started from the subsequent deposition of a 10-nm-thick $Al_2O_3$ etch stop layer and a 130-nm $SiO_2$ thin film onto a 200-mm-diameter glass substrate (Corning, 0.7 mm Eagle XG display grade glass). The subwavelength grating structures were then patterned by large-area deep ultraviolet (DUV) lithography followed by dry etching performed by a foundry (Moxtek, Inc. Orem, UT). The wafer was then diced into small chips (1.2 cm × 2.5 cm). A high-refractive-index $Si_3N_4$ film (~115 nm) and a thin $TiO_2$ biocompatible coating (~5 nm) were deposited onto the patterned substrate using a sputter (Kurt J. Lesker PVD 75) (see Supplementary for characterization details).

### Optical measurements

We used a home-built line-focusing microscope (Supplementary Fig. 3) to excite the PC–QD system and probe the QDs fluorescent intensity, emission spectrum, and scanning results of miRNA assays. A 450 nm laser diode (OSRAM, PL450B) serves as the excitation light source. The fluorescent signals are imaged by an imaging spectrometer (Horiba iHR 550) with an electron-multiplying charge-coupled device (EM-CCD) camera (Hamamatsu, C9100-13). All assay scanning images on PC and excitation enhancement factor tests are performed using a low NA ×50 lens and low laser power (Olympus LMPLFLN ×50, NA = 0.5; laser power = 1 mW). Since single QDs are not able to resolve on glass using a low NA lens, the extraction enhancement factor tests and blinking comparison experiments are performed using high NA ×100 lens and higher laser power (Zeiss alpha Plan-APO ×100 Oicl DIC, NA = 1.46; laser power = 2.1 mW). Experimental back focal plane images were taken by using a Bertrand lens (180 mm focal lengths) and ×100 Olympus objectives (LMPLFLN ×100, NA = 0.8).

Angle-resolved far-field characterization of bare PC was performed to demonstrate quality factors for calculating the estimated enhancement effects. The sample was covered in water and mounted on a motorized rotor with its $y$-axis aligned to the rotation axis. A collimated beam (area ~5 mm²) of white light from a Deuterium–Halogen lamp passes through a linear polarizer (TE polarization) and impinges the sample at incidence angle $\theta$. The zeroth-order reflected light at a series of incidence angles was collected using optical fibers and sent to the spectrometer (Ocean Optics 20000). The out-coupled emission spectrum of the PC–QD system was collected at an angle $\theta_{col} = 26°$ focused by a lens onto the slit of the spectrometer (Acton Research, SpectraPro 500i) with a CCD camera (Princeton Instruments, PIXIS)

Time-resolved photoluminescence (TRPL) was measured from ensemble QDs using a custom-built setup (Supplementary Fig. S4). The excitation source was a Ti:sapphire laser (Spectra-Physics Mai Tai) with an optical parametric oscillator, producing 100–120 fs pulses at $\lambda_{laser} = 425$ nm and a repetition rate of 80 MHz. The excitation was off-resonance from the PC pump mode ($\theta_{in} = 26°$) with a power of 2.1 mW. The emission was collected at a QD out-coupling angle ($\theta_{col} = 26°$) and passed through 550 nm long-pass filters to remove the excitation laser and imaged onto silicon single-photon avalanche photodiodes (Si-SP-APDs). The APDs were connected to a time-correlated single photon counting module, which assembled a single-photon parameter-tag mode and photon distribution modes to generate the histogram of photon arrival times. The temporal resolution of the system was ~22 ps.

All measurements were performed at an average power incident on the PC of 1–2.1 mW. Based on the dependence of emitted intensity as a function of excitation power, we conclude that measurements of all samples were carried out in the linear (unsaturated) regime (Supplementary Fig. S7).

### Electron microscopy characterizations

Transmission electron microscopy (TEM) images of the QDs were collected using the JEOL 2100 Cryo TEM operates at 200 kV. Scanning electron microscopy (SEM) images of the PC–QD sample were obtained using a Hitachi S-4800 field emission SEM. A thin conductive layer (~6 nm, gold–palladium for Bird's-eye view and carbon for cross-section view) was sputtered onto the sample before SEM.

### Atomic force microscopy (AFM) characterizations

The cross-sectional height profiles of the PC were analyzed with an atomic force microscope (AFM) (Cypher, Asylum Research) in contact mode using a monolithic silicon tip with a 30-nm-thick aluminum reflex coating (Budget Sensors).

### Photonic crystal (PC) surface functionalization with capture oligo

PC chips were sonicated in acetone (Sigma-Aldrich), isopropyl alcohol (Sigma-Aldrich), and deionized water respectively for 2 min and dried under a stream of compressed nitrogen, followed by a 200 W oxygen plasma treatment at a pressure of 500 mTorr for 10 min using a Pico Plasma System (Diener electronic, Germany). In a glass reaction chamber, (3-Glycidoxypropyl) trimethoxysilane (GLYMO, Gelest, Morrisville, PA, USA) was vapor-deposited on the PC surface in a vacuum oven at a temperature of 80 °C under 30 Torr for 4–5 h. For each PC chip in vapor deposition, 100 μL of GLYMO was added to the containing glass reaction chamber. The deposited PC chips were removed from the oven and sonicated in toluene (Sigma-Aldrich), methanol (Sigma-Aldrich), and deionized water, respectively, for 2 min, and nitrogen dried. For the DNA functionalization of a 1.25 cm² PC surface, a volume of 40 μL, 50 μM amino-terminated PC capture oligo (1×TE buffer, 0.05% TWEEN-20, pH = 9.0), and dispensed on the GLYMO-deposited PC surface. After 6 h of incubation at room temperature, the PC chips were rinsed by a gradual decrease of TE buffer concentration from 1× to 0.01×. A blocking buffer (SuperBlock TBS, Thermo Fisher Scientific) was added for 20 min and washed twice using 1× TE buffer before use.

### QD surface modification with probe DNA

Streptavidin (SA)-coated QD605 (Invitrogen) were separated from the supernatant after 3 min of centrifugation at 5000 × $g$. ssDNA probes and streptavidin (SA)-coated QD605 (Invitrogen) are conjugated at the stoichiometry ratio of 10:1 (Probe:QDs) in 1×TE buffer and incubated for 2 h at room temperature while mixing gently in a rotator to ensure sufficient reaction. The conjugate was then filtered using 0.5 mL Amicon filter (MWCO 50 kDa) to remove the free DNA probes from the solution. The final concentration of QDs is ~68 nM.

### Reporting summary

Further information on research design is available in the Nature Research Reporting Summary linked to this article.

## Data availability

The data that support the findings of this study are available within the article and its Supplementary Information, or from the corresponding author upon request.

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

## Acknowledgements

This work is supported by the National Institutes of Health (NIH) R01-GM108584 (B.T.C.), R01-CA227699 (A.M.S., B.T.C., and M.K.), R01-CA212097 (M.K.), National Science Foundation (NSF) CBET-1900277 (B.T.C.), the Carl Woese Institute for Genomic Biology postdoctoral fellowship (T.C and L.Z.), and the Cancer Center at Illinois for the C*STAR research fellowship (Y.X.) and TiME fellowship (O.H.A.). The authors gratefully acknowledge G.A. Fried, G. Popescu, J.A.N.T. Soares, P.R. Selvin, S. Sarkar, O.R. Teniola, D.E. Vaz, L.E. Kwon, J. Tibbs, S. Shepherd, B.K. Maity, A.J. Cyphersmith, E. Araud, C.-W. Kuo, Y. Zhuo, P. Le, F.-C. Hsiao, L.D. Akin, S. Nalla, N. Chauhan, A. Igarashi, and the rest of Nanosensor group for valuable discussions. The authors thank H. Zhou, J.C. Spear, K.A. Walsh, W. Swiech, and K.M. Flatt from Materials Research Lab at the University of Illinois at Urbana-Champaign for their training and support, and W. Wang for the assistance in coverslip cleaning. The authors also acknowledge MOXTEK, Inc. and Y. Yan for providing the SEM images of the grating structure.

## Author contributions

Y.X. and B.T.C. conceived and designed all the experiments. Y.X. performed all the optical experiments, bioassay experiments, built the PCEF optical setup with assistance from Q.H., C.M.R., Xing W., and L.Z. Y.X., Q.H., A.M.S., L.Z. S.L. Xiaojing W., and T.D.C. interpreted the data. C.M.R. designed the PC structure and line-scanning software. Y.X. and T.D.C. designed the bridge assay. Y.X., P.B., S.L., and Q.H. performed the simulations. S.L., Y.X., and P.B. calculated the enhancement factor. O.H.A. performed the gel electrophoresis. C.C.

performed the diffusion simulation. Y.X. performed SEM, TEM, AFM, and optical characterization. L.Z. and Xing W. provided experimental materials for bioassay in human serum. B.T.C., A.M.S., M.K., and Xing W. supervised the study. M.K. identified the miRNA target biomarker. Y.X. wrote the manuscript with input from all authors.

## Competing interests

The authors declare no competing interests.
