## [Peer Review File · Nature Communications]

Photonic Crystal Enhanced Fluorescence Emission and Blinking Suppression for Single Quantum Dot Digital Resolution BiosensingREVIEWER COMMENTS

Reviewer #1 (Remarks to the Author):

The manuscript by Xiong et al. reports a thorough and impressive study of the use of enhanced quantum dot emission from a grating surface for implementing a digital miRNA assay. The improvements afforded by the optical enhancements are cleverly leveraged to make significant improvements to conventional microscope assays. In particular, the fact that only the quantum dots that bind to the miRNA target as part of the assay sandwich become bright, is a very elegant way to deal with potential background signal from unbound dots. The descriptions and figures are very detailed and comprehensive and support the narrative very well. The manuscript is suitable for publication in Nature Communications if the authors can address the following issues:

- The analysis of the optical improvement in the presence of the photonic crystal grating is very clear and well described. However, the effect seems to be essentially identical to what the group reported in a seminal paper in Nature Nanotechnology over a decade ago. It would be helpful if the authors could clarify if and how the present demonstration differs from their own work (refs. 27-29).
- The assay appears to be a sandwich assay. It may be simpler to just call it that.
- The authors did carry out simulations for a negative control experiment with non-matching miRNAs. However, the manuscript would have been significantly strengthened if a negative control experiment were reported and if the miRNAs were detected from a more meaningful (native) sample matrix. Were any of these carried out?
- While it may be correct that TIRF microscopy has a dynamic range of 5 logs, there have been numerous reports of lab-on-chip based digital assays with significant larger dynamic range. Providing better context for the present work might be helpful.
- Fig. 4b reports an impressive performance of the sensor 9 orders of magnitude in concentration. However, the number of detected quantum dots only increases by about a factor of 10, suggesting an extremely nonlinear performance with very little wiggle room for distinguishing vastly different concentrations. Shouldn't the assay be linear? The authors should provide a discussion of this nonlinearity. Can the observed relationship between number of targets and detected number of quantum dots be modeled?
- The sensor has a relatively long incubation time of 2 hours. How was that chosen? Can it be realistically improved? Can the authors add a figure that shows the performance of the sensor (# of detected QDs) versus incubation time?
- Related to the previous issue, how effectively are the QD probes and target molecules transported to the sensor surface? Can the authors provide more details on how the complete sensor looks like in terms of amount of fluid probed, dimensions of the compartment etc.? An image of the complete device (perhaps in the supplemental information) would help understand the full system arrangement.

Reviewer #2 (Remarks to the Author):

In this manuscript, the authors demonstrated that the photonic crystal can greatly enhanced the excitation, directional extraction, and blinking suppression for single quantum dot digital resolution biosensing. It is an attempt to gain more control over photonic crystal enhancement of quantum dot luminescence and is thus of interest to the broad research community of nanophotonics and biosensing. The paper is recommended for publication after addressing the comments below.

1. On page 9, the authors state that high concentration drop-casted QDs (1 μ M) spread uniformly on the PC surface without aggregation (Figure 1a). Does this mean a monolayer of QD was formed on PC surface? A SEM image of a small area (500 nm x 300 nm) is not sufficient to confirm the uniformity. A statistical analysis of multiple images will be helpful to confirm this. There is a concern that the density of QDs is heterogeneous due to the coffee ring effect associated with drop-casting preparation.

2. What is the size uniformity of QDs? What is the distance between QDs and the PC surface? How does the distance/size uniformity of QDs affect the enhancement?

On page 13 line 270, the enhancement factor is calculated based on the assumption of a 4 nm thick QD layer above the PC grating surface. How does the location (ridge or groove) and density of QDs affect this enhancement?

3. On page 24 line 496, the authors state that without the target miRNA, the QDs will not be pulled down to the PC surface and will not experience 3000X enhance. What is the EM decay length of PC? If free QDs are present near the PC surface, will the enhancement the same? A control experiment needs to be performed to confirm this.

4. In Fig. 1e and Fig.S3, the authors show their experimental setup, in which the sample is illuminated at normal incidence and the resultant PL was measured. This is acceptable only when the luminescence pattern (meaning PL intensity as a function of emission angle) remains unchanged from sample to sample. The PC can exhibit strongly modified luminescence pattern due to diffraction. Therefore, the PL should be measured using an integrating sphere. If this is not possible, the authors should at least provide the PL intensity at various angles and show that the measured photon intensity enhancement has properly taken into account any variations in luminescence pattern.

5. On page 25 line 507, the authors state that if the number of bases of the capture changes from 10 to 7, the capture forms a "hairpin loop" structure and has no ability to pair with the bottom part of the target miRNA. This is surprising because a longer sequence typically tends to fold but the authors postulate otherwise. A short one probably still binds to the target with less affinity. Again, control experiments to support this postulation are needed.

6. In the miRNA Bridge Assay application, the authors used the oligo pairs (probe-target) design to ensure the specificity. Some information provided is not consistent. Figure 3b shows the schematic of double hybridization, which is not consistent with the sequence description in the supporting information. The information is critical for data interpretation of kinetic results described in Figure 4c-4i. Following the 3' and 5' described in Figure 3b, the mutated base in the target miRNA is located in the region which interacts with the capture. It is puzzling why this induces the kinetic behavior of probe-target interactions. It seems that the interpretation of the kinetic data is based on the mismatch is between the probe and target shown in the Figure S8c. Did these experiments involve washing? In some experiments, each incubation step is accompanied by a gentle washing step to remove the free targets and unbound QDs was described on page 25 line 521. However, the ability of discriminating a single base difference in a target miRNA in 10 minutes without a washing step was described in the introduction on page 7 line 158. Detailed experimental description is needed to support such claims since washing procedures are generally important in biosensing experiments.

7. On page 28 line 569, the authors state that three independent double-blind experiments have been carried out. However, no discussion on the statistical analysis of the results.

8. Multiple grammar mistakes need to be corrected during the revision.

Reviewer #3 (Remarks to the Author):

The paper is interesting, providing an ultrasensitive platform for biomedical analysis. It comprises a wealth of data on the analysis of single QD particles for high sensitivity measurements using photonic crystals. The questions that I have are both general, around the structure of the paper, presentation and novelty, as well as specific questions relating to key data.

General Comments

1. Firstly, there is a general question as to the value of this technology, analytically. With sample preparation taking several hours and measurement then taking an addition 2 hours, how valuable is this to clinical practice. The authors should more clearly articulate a vision as to why this is important to the wider readership of Nature Comm (put another way, what is the clinical decision and care pathway that will be critically affected by whether there is 10aM or 100aM of miRNA in a sample?)

2. I would also argue that the introduction is not written to identify novelty. There are a lot of references cited for every part of the system, so the authors are indicating that the only novelty is in the integration. The authors have to clearly show that this is more than simply the sum of the parts,

which is not at all clear in the writing in the current form.

3. On the subject of (2), the claim of a X3000 fold enhancement is relative to a glass slide as reference measurement. What is more interesting to me as a reviewer and indeed to the readership is what the improvement is, relative to other nanostructured photonic systems, especially those using photonic crystals. Again, a detailed critical view is needed.

4. The paper would also greatly benefit from re-organisation of the figures. The multiple panels serve only to confuse the reader - there is a strong case for some of these to be moved to ESI (especially in Fig 1., as the panels do not evidence the major point of the paper). I think the authors need to make decisions on key data that corroborates the message(s) in the paper. I believe the paper would be more understandable if Figure 3 was actually Figure 1 and was referenced in the introduction.

Specific Details

Most of my specific comments are focussed around Figure 4 – which I believe is the key result. As advice, by combining both static and dynamic measurements into a single panel this obscures and confuses – this data should sit within 2 separate figures, associated with the two sections in the text. This is the key experimental evidence and readers must be allowed to understand it.

1. Figure 4a has different scaling in inset is also difficult to interpret. Both images at the same scale should at least be provided in Supplementary Information, with the same resolution. Rather than see all 9 concentrations (+ background) better to show less at the same resolution to allow for a critical comparison.

2. Figure 4b is too difficult to understand with the double axis with different scales. I suspect the reference should also be the background unmodified glass (grey), not the noise (blank +3sd, blue), so that the authors show how much better their device performs relative to an unstructured glass slide. The signal to background would be 3 sd above the grey, not blue, line.

3. Fig 4 b, Why did the experiment stop at 10 aM? Given the linear plot over 9 orders of magnitude concentration, and that the signal should still be apparent above the background at 2 orders of magnitude below the current LOD, at 100zM, why did the authors not extend this investigation. If the linear relationship does indeed extend at lower concentrations against the background, this should be shown. Similarly if it does not, it should still be shown so as to give a better understanding of the system.

The same is true at the high concentrations to provide a complete understanding of the technique. Why are measurement above and below the linear range not shown.

4. Figure 4c-d are unclear (the positioning of the particles looks exactly the same and the cross covers a lot more than a single base). The Figure certainly does not give credit to the 22 different possibilities that the authors have analysed.

As a final comments, some thought about significant figures and errors – e.g 1.708ns (+/-0.3nS). Does this measurement really have a precision around 1 ps or is this algorithmic from the fitting function used in Fig 2(g), and what does that value mean when the error is 2 orders of magnitude higher than the measurement. Do the authors mean 1.7 ns (+/-0.3nS) which is less convincing, or is this a typographic error e.g. 1.708ns (+/-0.3pS).

Similarly in Figure 4(i), the reader should have a view of variability (how do the 5 trajectory measurements compare)? Another example of important information almost lost by the compression of 9 figures into one panel, and by merging the static and dynamic data in the same figure.

I would like if possible to see a revised manuscript.

Reviewer #1 (Remarks to the Author):

The manuscript by Xiong et al. reports a thorough and impressive study of the use of enhanced quantum dot emission from a grating surface for implementing a digital miRNA assay. The improvements afforded by the optical enhancements are cleverly leveraged to make significant improvements to conventional microscope assays. In particular, the fact that only the quantum dots that bind to the miRNA target as part of the assay sandwich become bright, is a very elegant way to deal with potential background signals from unbound dots. The descriptions and figures are very detailed and comprehensive and support the narrative very well. The manuscript is suitable for publication in Nature Communications if the authors can address the following issues:

Our reply:

We thank the reviewer for their enthusiastic comment!

Reviewer Comment:

The analysis of the optical improvement in the presence of the photonic crystal grating is very clear and well described. However, the effect seems to be essentially identical to what the group reported in a seminal paper in Nature Nanotechnology over a decade ago. It would be helpful if the authors could clarify if and how the present demonstration differs from their own work (refs. 27-29).

Our reply:

The results reported in this manuscript represent **seven** fundamentally important advances from the 2009 *Nature Nanotechnology* paper in the demonstration of novel physics principles, newly designed photonic crystal structure, and the compelling new bio-applications enabled by them.

(a) The biggest differentiating feature between the present work and our 2009 report is the greatly improved enhancement factor. We here experimentally demonstrated (also supported by numerical simulation) a **~3000x** enhancement in signal for QDs on a photonic crystal (PC) surface, while the previous work demonstrated a **108x** enhancement.

This much greater enhancement is achieved through precise tuning between radiative quality factor and non-radiative (absorption or inevitable loss cause by fabrication imperfections) quality factor in two PC's resonance channels (pump-mode and fluorescence-mode) through the Q-matching condition. As a result, we maximize the enhancement factors close to its theoretical limit. The 3000x enhancement is also attributed to innovative integration of five multiplicative enhancement factors to provide enhancement all the way from the fluorescence generation process (absorption, non-radiative transition/excited-state lifetime, radiative fluorescence emission) to the collection process (far-field distribution and blinking reduction) after the emitted photons are generated. Such enhancement effects include (1) enhanced laser excitation field to increase QD absorption rate, (2) increased quantum efficiency through Purcell effects to minimize the non-radiative transition lifetime, (3) enhanced photon extraction rate during photon generation, (4) greater photon collection efficiency through PC guided directional emission, and (5) blinking suppression to reduce the "off" time. The 2009 *Nature Nanotechnology* paper was only focused on achieving the enhancement through the enhanced photon extraction effect. Without optimized quality factors and field distributions, the previous PC design reported in our 2009 *Nature Nanotechnology* paper

did not achieve lifetime enhancement and collection efficiency improvement. Nevertheless, without single QD detection capability, the previous work **could not address** the PC effects in terms of QD blinking modification.

(b) Further, the simplified photonic crystal structure design used in the present manuscript helps to maximize all the enhancement factors. While the 2009 paper utilized a 2-dimensional PC design that requires a perfectly arranged square lattice hole array, its enhancement factor was mainly restricted by mismatch between high radiative quality factor design and fabrication imperfections (low nonradiative quality factor Q_{nr}). In our improved structure design, we were able to achieve much greater enhancement effects using a much simpler but well-optimized 1-dimensional PC grating which depends less on precise fabrication tolerances and achieves moderate Q_{nr} . The simplified PC structure design is needed to achieve the highest enhancement, since a photonic system will reach its maximum enhancement under Q-matching conditions where the quality factors of radiative decay Q_r are similar to the nonradiative Q_{nr} decay.

(c) Another important differentiating feature is the low-cost fabrication process. The 1-dimensional PC used in this work was made by a large-area holographic lithography fabrication process on an 8-inch diameter glass plate that is amenable to low-cost manufacturing ($\sim \$0.2/\text{mm}^2$), while the 2-dimensional PC used in the 2009 study was a much smaller (3 mm^2) PC fabricated by electron beam lithography with a lengthy fabrication time ($>20 \text{ hours}/\text{mm}^2$) and high cost ($\sim \$2500/\text{mm}^2$). Thus, the inexpensive large-area PC fabrication is an important element of a device to be used for molecular diagnostics applications.

(d) Additionally, we showed the ability to observe individual QDs with high signal-to-noise ratio (~ 60), even with air-based low NA (0.5) microscope objectives as an important practical consequence of the greater enhancement offered in our current work. On the contrary, the 2009 study only described “ensemble” images of a drop-cast QD-layers ($>1 \text{ mM}$) on the PC surface, without the ability to detect individual QDs. Our newly-demonstrated enhanced signal/noise offers the ability to achieve single QD counting-based assays and the ability to study blinking fluctuations of QD photoluminescence without expensive optical components.

(e) Furthermore, by utilizing the “digital counting” capability of individual QDs, we performed ultrasensitive biosensing assays, where individual QD tags represent the presence of only one target biomolecule from a test sample. In this novel study, we demonstrate a simple assay for the detection of a cancer-specific miRNA molecule, with a limit of detection of only ~ 10 attomolar (aM). The 2009 study did not describe any biodetection assays.

(f) Nevertheless, with the advancement in single QD detection ability, we are able to study the QD photoluminescence phenomenon change when it on all-dielectric PC surface. To our knowledge, this is **the first time that QD blinking suppression being reported** in such photonic system. This novel capability **improves the QDs “on-time” from 15% to 85%**, providing an innovative method to ameliorate signal intermittency issues encountered during ultrasensitive measurements and fast motion tracking at the single particle level. Additionally, by utilizing the blinking suppression capability motion in single particle tracking, we clearly observe differential surface motion trajectories of individual QDs when their surface attachment stringency is altered by changing a single base in the target miRNA sequence.

To clarify these points for the reader, we have added the following text to the beginning of the third paragraph in the Introduction:

(Page 4, Line 93): Advancing beyond earlier reports of quantum dot (QD) enhancement by PC surfaces¹, our present goal is to construct a newly designed PC nanostructured surface that can serve as a general-purpose macroscopic substrate for fluorescence microscopy of QD tags with a simple low-cost fabrication process and greatly improved enhancement factor. The much greater enhancement is achieved through radiative engineering, where precise tuning of two quality factors of radiation (Q_r) and non-radiation (absorption or inevitable loss cause by fabrication imperfections, Q_{nr}) to match each other. Q-matching requirements ($Q_r = Q_{nr}$) need to be satisfied for two PC resonance modes (pump-mode and fluorescence-mode), in order to maximize the enhancement factor toward theoretical limits. The 3000x enhancement is also attributed to integration of five multiplicative enhancement factors to provide comprehensive enhancement all the way from the fluorescence generation process (absorption, non-radiative transition/excited-state lifetime, radiative fluorescence emission) to the collection process (far-field distribution and blinking reduction) after the photons are generated. As a result, we increase the signal-to-noise ratio (S/N) and suppress fluorescence intermittency for individual QD imaging...

(Page 5, Line 122): ...A key novel element of the current study resides in the combination of several independent multiplicative effects that utilize a PC surface to achieve an unprecedented 3000x enhancement in QD emission with blinking suppression, which enables observation of individual QDs using a low NA objective, while maintaining a high signal-to-noise ratio, a large field-of-view, and improvement of signal intermittency issues.

(Page 5, Line 131): ...A further novel aspect of this study is to utilize the single-QD imaging capability in a digital-resolution biomolecular assay that has achieved ultrasensitive and ultraspecific detection of a miRNA biomarker which can be adapted to detect other miRNAs as well as DNAs and proteins...

Reviewer Comment:

The assay appears to be a sandwich assay. It may be simpler to just call it that.

Our reply:

We thank the reviewer and we have carefully considered this suggestion. The term “sandwich assay” is generally used to describe protein-protein detection, in which an antigen (the analyte) is sandwiched between a capture antibody and a detection antibody that recognize separate epitopes of the antigen. In discussion with our colleagues in the field of nucleic acids biosensing, we prefer to remain with the term “hybridization” assay, as our nucleic acid target forms two separate base pairing regions respectively with the PC surface-immobilized ssDNA capture and the ssDNA probe on the quantum dot. If the reviewer does not strongly object, we feel that the term “hybridization” will be most conventional representative of nucleic acids base-pairing interactions, as the target miRNA forms a “bridge” that hybridizes across the upper/lower ssDNA probes.

Reviewer Comment:

The authors did carry out simulations for a negative control experiment with non-matching miRNAs. However, the manuscript would have been significantly strengthened if a negative control experiment were reported and if the miRNAs were detected from a more meaningful (native) sample matrix. Were any of these carried out?

Our reply:

We thank the reviewer for the insightful comment! Indeed, showing the detection of miRNAs in a biologically representative sample matrix is very important and can strengthen the value of our biosensing platform. Therefore, we carried out all the assay experiments to detect miRNAs in an unprocessed human serum. Additionally, we evaluated the selectivity of the assay method using three miRNAs with clinically relevant mutations that each carries a single base mismatch at a different position of target molecule to show the high detection specificity of our engineered ssDNA capture and probe oligos.

The following modification was made to the revised manuscript:

(Page 24, Line 601): Circulating miRNAs are highly stable in human serum², and the direct detection of miR-375 in serum was recently shown possible³. To demonstrate the feasibility of our QD-based digital sensing in an unprocessed native sample matrix, we spiked miR-375 targets in human serum (Sigma-Aldrich, H3667) at 12 final concentrations ranging from 100 zM to 10 nM, and then carried out the sensing assays without performing any RNA extraction or purification.

Further clarification of the probe design for selective detection of miRNA target was added to the Supplementary Information. We used NUPACK to predict the binding efficiency of all 22 mismatch variations along the miR-375 target (Fig.S8), showing that our designed “bridge” assay could be highly selective against a single nucleotide mismatch at #4, #5, #6, #7, or #8 base on miR-375 (counted from the 5' end), where this specific mutation sits which has been reported to have a clinical significance⁴. In the original manuscript, we experimentally performed single-base variation test only upon index #4 (1MM4) in buffer media.

As suggested by the reviewer, we carried out more selectivity experiments to show that our sensor can distinguish a single nucleotide mutation placed at the #4, # 6 or # 8 position of the control miRNA at 100 pM concentration in human serum (compared with the perfect-match target miRNA at same concentration).

The following modification was made to the revised manuscript:

(Page 23, Line 594, Figure legend): **(e)** Single-base mismatch discrimination test. Line-scanning image panel demonstrates digital resolution of captured QDs for target miR375 perfect match group (PM), versus three different SNVs at 100 pM. **(f)** Quantification of perfect match target sequence and the SNV cases at 100 pM concentration. Statistical significance was tested using independent t tests; ****P < 0.0001. The error bars represent the standard deviation of three independent assays.

(Page 25, Line 640): Figs. 4e-f illustrate a dramatic decrease in particle counting, resulting from SNVs at index #4, #6, and #8, with a range of 90-92% signal reduction (difference in counting results: $\Delta N = \frac{N_{PM} - N_{MM}}{N_{PM}} = 90-92\%$, with two-tailed p-values of <0.0001) compared with the perfect-match group (PM) at 2 h.

Reviewer Comment:

While it may be correct that TIRF microscopy has a dynamic range of 5 logs, there have been numerous reports of **lab-on-chip based digital assays** with significantly larger dynamic range. **Providing better context** for the present work might be helpful.

Our reply:

We agree with the reviewer's assessment. Searching the literature for reports of lab-on-chip based non-amplification digital-resolution assays with high dynamic range using direct-counting, the report with the greatest range is from our own group but using a different biosensing detection method. We have now incorporated the reference⁵ in the revised manuscript to provide better context for digital counting assays using lab-on-chip system.

To provide better examples in response to the reviewer's comment, we also include three more references that reported amplification-based digital PCR assays using lab-on-chip system. The following modification was made to the revised manuscript to provide a better context of existing technologies:

(Page 21, Line 569): Conventional surface capture assays can measure fluorescently labeled analytes across a 1000-fold concentration range and at the sub-nanomolar level, but many biological molecules exhibit more than 1,000,000-fold variations in abundance down to the sub-femtomolar level. **Lab-on-chip digital PCR amplification-based assay can offer absolute quantification for nucleic acid with an improved dynamic range up to 5 orders of magnitude⁶⁰⁻⁶² with precise thermal cycle control. Without enzymatic amplification, our previous research⁶³ combine single-molecule counting and intensity calibration showing the dynamic range of fluorescence assays can be expanded to similar range (10⁵-fold) and reach ~10fM detection limit using a TIRF microscope with a NA=1.46 100x lens and high-gain EMCCD. Ultimately, we demonstrate the multi-enhancement from the PC-QD assay can provide enzyme-free digital resolution biosensing ability for direct counting single miRNA with inexpensive optics (NA=0.5 objective, low power laser diode, and low EM gain CCD) ...**

Reviewer Comment:

Fig. 4b reports an impressive performance of the sensor 9 orders of magnitude in concentration. However, the number of detected quantum dots only increases by about a factor of 10, suggesting an extremely nonlinear performance with very little wiggle room for distinguishing vastly different concentrations. Shouldn't the assay be linear? The authors should provide a discussion of this nonlinearity.

Our reply:

We realized that plotting two dose response curves for both digital counts and analog intensity, in the same graph, by using double y-axis is a source of confusion. Thus, we have replotted them in two separate figures in the revised manuscript. The PC-enhanced QD digital counting results (left) are shown on Fig 4.d in main text. The plot on the right (our comparison for performing the same assay on a glass surface by using analog fluorescent intensity measurements) is referenced in the main text of the manuscript, and appears in the Supplementary Information (Fig S14).

As shown above, when plotting the dose response curve (10 aM-1 nM) for PC-enabled digital counting results in a log-log plot, the curve is a straight line which indicates the linear relationship between $Y = \log(y)$ and $X = \log(x)$. Here, as a digital counting biosensor, the definition of “linear-dose response” is different from the strict linear potential relationship. The constraints of linear response here are whether a linear relationship can be preserved when both axes are transformed by the same function.

The log-log “linear” response of our sensor could be caused by (a) the diffusion limits of the target especially at lower concentrations, and (b) the available binding sites on the PC surface were being occupied at high concentrations and thus, making it more difficult for new particles to bind as the concentration increased. However, the established calibration curves (Fig. 4b) demonstrate no significant overlap between adjacent concentrations based on the standard deviation values.

Reviewer Comment:

Can the observed relationship between number of targets and detected number of quantum dots be modeled?

Our reply:

Depending on the bioassay result, different statistical models can be employed to fit the data to generate a typical standard curve. Two-step biological assays involving DNA-RNA interactions usually do not follow a linearly proportional dose-response relationship. Instead, the dose-response of such assays that cover a full range of analyte concentrations often yields a four (or five) parameter logistic model with only a middle portion showing log-log “linear” response⁶.

Following the reviewer’s suggestion, we expanded the range of analyte concentrations (100 zM to 10 nM) and performed additional experiments to examine the sensor response in triplicate in the human serum media. We tested four different statistical models that are commonly used in multi-step biological assays to model our sensor response, including log-log “linear” model, quadratic model (quadratic polynomial), four parameter logistic model, and five parameter logistic model.

- 1) Log-log “linear” model:

$$Response = a + b * (concentration) + error$$

Where a and b are intercept and slope respectively. This model is usually employed after log transformation of the concentration and signal response and therefore is also termed as “log-log linear model”.

- 2) Quadratic model (quadratic polynomial)

$$Response = a + b * (concentration) + c * (concentration)^2 + error$$

Where a, b, and c represent intercept, linear, and quadratic term coefficients, respectively.

- 3) Four parameter logistic model

$$Response = top + \frac{bottom - top}{1 + \left(\frac{concentration}{EC_{50}}\right)^{slope}}$$

Where top and bottom represent the top and bottom asymptotes of the curve. EC₅₀ refers to the analyte concentration at which the signal response is halfway between top and bottom values. Slope represents the slope of the response curve at EC₅₀.

- 4) Five parameter logistic model

$$Response = top + \frac{bottom - top}{\left[1 + \left(\frac{concentration}{EC_{50}}\right)^{slope}\right]^{asymmetry}}$$

Where a fifth parameter, asymmetry, is introduced in the model, which indicates the degree of asymmetry of the “S-shaped” sigmoidal curve with respect to “EC₅₀”.

We have accordingly updated the results in SI.Fig S.12 with the following description:

(SI-Page 16, Line 270): “Based on the r-squared values, our results show that a full range (extended 12-order) of analyte concentrations can be best fitted by a five-parameter logistic model while the middle 9-order of concentrations (10 aM-1 nM) can be well fitted with a log-log “linear” model.”

Supplementary Figure S12. Calibration curves of the miR-375 detection assay in human serum. For the full concentration range from 100 zM to 10 nM, a five-parameter logistic model:

$y = top + \frac{bottom-top}{\left[1 + \left(\frac{x}{EC_{50}}\right)^{slope}\right]^{asymmetry}}$ was used to fit the dose response, where x is the concentration

of the targeted miRNA molecules, y is the QD counts. The five-parameter logistic fitting equation was $y = 537.8 + \frac{2.75-537.8}{\left[1 + \left(\frac{x}{43297}\right)^{2.162}\right]^{0.081}}$ with $R^2 = 0.9886$ (the green dash line). For the middle

concentration range from 10 aM to 1 nM, a log-log linear model: $\log(y) = a \cdot \log(x) + b$ was used to portrait the dose response. The log-log linear regression fitting equation is $\log(y) = 0.1692 \cdot \log(x) + 1.647$ with $R^2 = 0.9825$ (the yellow line). The error bars represent the standard deviation of three independent assays.

Reviewer Comment:

The sensor has a relatively long incubation time of 2 hours. How was that chosen? Can it be realistically improved? Can the authors add a figure that shows the performance of the sensor (# of detected QDs) versus incubation time?

Our reply:

We have performed additional experiments to characterize the assay response as a function of QD probe incubation time (8 time points from 30 mins to 4 hours) after exposure to the immobilized target miRNA on the PC. Our results, as plotted in Supplementary Fig.S14, confirm that QD

counting shows no significant change after 2 hours. The 2-hour incubation time is determined by the sensing environment in which analytes in a stagnant liquid volume in a well is limited by diffusion. The diffusion factor is thoroughly described in ⁷, in which the flux of analytes to the capture surface decreases with time, as the analyte concentration in the bulk solution is reduced by the capture process. The following discussion was included in the Supplementary Information accompanying Figure S14:

(SI-Page 17, Line 286):

Supplementary Figure S14. The quantification of the detected QDs versus incubation time for four different target concentrations. Each data point represents the average of 3 independent experiments. Error bars represent the standard errors.

Reviewer Comment:

Related to the previous issue, how effectively are the QD probes and target molecules transported to the sensor surface? Can the authors provide more details on how the complete sensor looks like in terms of amount of fluid probed, dimensions of the compartment etc.? An image of the complete device (perhaps in the supplemental information) would help understand the full system arrangement.

Our reply:

The detail study for the transport of target molecules and nanoparticle probes to PC surface-based sensor embedded in PDMS cartridges have been reported in previous publications with COMSOL model simulation as referenced in ⁵. At the concentration utilized in our assay, the QD probes can transport to the PC surface in 5-10 minutes. Meanwhile, target molecules (particularly at low concentrations) should reach equilibrium in approximately 2 hours. A schematic drawing of the PC biosensor with an applied array of PDMS wells is shown in Supplementary Information Figure S16, and a photo of the actual PC sensor next to a coin is included. The dimensions of the well (1.5mm diameter) and its volume (45 ml) are now included in the main text and in the figure

caption in SI Figure S16.

(SI-Page 19, Line 301):

Supplementary Figure S16. PC biosensor system. (a) Schematic drawing of the PC biosensor with applied array of PDMS wells. Well diameter: 1.5 mm; Sample volume: 45 mL. (b) A photo of the actual diced PC next to a coin. Dimensions: 12 mm×25 mm×0.8 mm.

Reviewer #2 (Remarks to the Author):

In this manuscript, the authors demonstrated that the photonic crystal can greatly enhanced the excitation, directional extraction, and blinking suppression for single quantum dot digital resolution biosensing. It is an attempt to gain more control over photonic crystal enhancement of quantum dot luminescence and is thus of interest to the broad research community of nanophotonics and biosensing. The paper is recommended for publication after addressing the comments below.

Our reply:

We thank the reviewer for the complement of our work!

Reviewer Comment:

On page 9, the authors state that high concentration drop-casted QDs (1 μ M) spread uniformly on the PC surface without aggregation (Figure 1a). Does this mean a monolayer of QD was formed on PC surface? A SEM image of a small area (500 nm x 300 nm) is not sufficient to confirm the uniformity. A statistical analysis of multiple images will be helpful to confirm this. There is a concern that the density of QDs is heterogeneous due to the coffee ring effect associated with drop-casting preparation.

Our reply:

When preparing the SEM sample, we minimized coffee ring effect by applying a droplet of QD solution to the PC surface for only a few seconds then immediately removing the droplet without letting the QD sample evaporate to dry. We then washed out the excess QDs with DI water and kept the sample in desiccator overnight before SEM imaging. An example larger area SEM image is shown below after such sample preparation:

More importantly, we thank the reviewer for pointing out the surface-based SEM image is not a best way to examine QD aggregation sufficiently. Thus, we removed the comments on the assessment of QD aggregation by using SEM. Instead, we only use the SEM image to demonstrate the concept of PC+QD system in Fig 2a (page 8, line 309).

The following modification was made to the revised manuscript:

(Page 8, line 219): Figure 2a shows a representative scanning electron microscope (SEM) image of the PC-QD system ~~where the high concentration drop-casted QDs (1 μ M) spread uniformly on the PC top surface without aggregation.~~

Additionally, instead of using SEM, we used Dynamic Light Scattering (DLS) for statistical analysis and prove that there is no aggregation of our QDs. The data appears in Supplementary Information Figure S6

(SI-Page 10, line 166):

Supplementary Figure S6. Hydrodynamic size distribution of QD605. The average hydrodynamic diameter measured by DLS is 20.65 nm with a standard deviation of 1.295nm.

Reviewer Comment:

What is the size uniformity of QDs?

Our reply:

In addition to DLS measurements, which show a hydrodynamic QD diameter distribution of 20.65 nm, we have also collected multiple TEM images of our commercially purchased QDs. We have included two representative TEM images with size distribution analysis (# of QDs > 400) as Supplementary Information Figure S7, shown below. The particle size distribution analysis shows the average diameter of QD605 with complete core-shell structure is 7.94 ± 0.83 nm.

(SI-Page 10, line 170):

Supplementary Figure S7. QD605 particle hard-size distribution analysis by TEM imaging. Representative TEM images for commercially purchased QDs at (a) 500K and (b) 100K magnifications. (c) Practical size distribution analysis of QD605. The average diameter of QD605 with complete core-shell structure is 7.94 ± 0.59 nm (number of analyzed QDs > 400).

Reviewer Comment:

What is the distance between QDs and the PC surface? How does the distance/size uniformity of QDs affect the enhancement? On page 13 line 270, the enhancement factor is calculated based on the assumption of a 4 nm thick QD layer above the PC grating surface.

Our reply:

The distance between center of QDs and PC surface is estimated as 4 nm, as the average diameter of QDs is about 8nm. Based on reviewer's suggestion, we performed additional numerical and

theoretical analysis on Z-distance dependence of total enhancement factor and added a section in SI. Part 11. “Z-distance dependence of total enhancement factor”.

(SI-Page 11, line 188): We investigated the Z-distance dependent total enhancement factor with the same simulation model but run a series of test by parameter sweeping in the QD Z-height as a variable d . The simulated enhancement factor Λ (normalized to the 4-nm result) is shown in the Fig. S8 (b). As expected, the greater the distance between QDs and the PC surface, the smaller the electric field experienced by the QD, which leads to a z-dependent enhancement factor. The plot shows that QD enhancement factor decreases approximately linearly as the QD is moving away from the PC surface for the first ~20 nm in z-height, and that within our anticipated region of QD displacement of 0-10 nm for our miRNA assay, the QDs will be between 100-88% of their maximal enhancement. When QDs are more than 60nm away from the surface, there will be almost no enhancement effects.

Supplementary Figure S8. Z-distance dependence of total enhancement factor. (a) Schematic illustration of QD layer and integration region. (b) Simulated distance dependence of the enhancement factor with fitting. A linear model: $y = a + bx$ was used to fit for distance smaller than 20nm, where y is the normalized total enhancement and x is the distance above PC surface. The linear fitting equation was $y = 1.066 + -0.0183x$ with $R^2 = 0.997$ (the red dash line). A quadratic model: $y = 1.079 + -0.0214x + 0.00015 x^2$ was used to fit when d between 4 to 60nm with $R^2 = 0.999$ (the yellow line).

We here provide a simple but accurate explanation to the linear decrease with small d . We take the excitation enhancement factor as an example. $\Lambda_{excitation}$ is proportional to α^P / d , i.e., electromagnetic energy of pump-mode stored in the QD layer and the evanescent electric field can

be written as $\sum_{\mathbf{G}} E_{\mathbf{G}}^P e^{-z/z_G^P} e^{i(\mathbf{k}_{\parallel} + \mathbf{G}) \cdot \mathbf{r}}$, by using the Bloch theorem. Here k_{\parallel} is the in-plane wavevector, \mathbf{G} is any of the reciprocal vectors and $z_G^P = 1 / \sqrt{(\omega / nc)^2 + |\mathbf{k}_{\parallel} + \mathbf{G}|^2}$ is the corresponding evanescent tail length. Thus, we have

$$\Lambda_{excitation} \propto \alpha^P / d \propto \int_{QD_layer} \left| \sum_{\mathbf{G}} E_{\mathbf{G}}^P e^{-z/z_G^P} e^{i(\mathbf{k}_{\parallel} + \mathbf{G}) \cdot \mathbf{r}} \right|^2 dx dy dz / d$$

$$\propto \sum_{\mathbf{G}} |E_{\mathbf{G}}^P|^2 \int_0^d e^{-2z/z_G^P} dz = \sum_{\mathbf{G}} |E_{\mathbf{G}}^P|^2 \frac{z_G^P}{2d} (1 - e^{-2d/z_G^P}) \sim \sum_{\mathbf{G}} |E_{\mathbf{G}}^P|^2 (1 - (d / z_G^P)) = 1 - \frac{\sum_{\mathbf{G}} |E_{\mathbf{G}}^P|^2 / z_G^P}{\sum_{\mathbf{G}} |E_{\mathbf{G}}^P|^2} d,$$

which is linear with small d , and same for extraction enhancement factor. The total enhancement can be determined as

$$\Lambda \propto \Lambda_{excitation} \Lambda_{extraction} \sim \left(1 - \frac{\sum_{\mathbf{G}} |E_{\mathbf{G}}^P|^2 / z_G^P}{\sum_{\mathbf{G}} |E_{\mathbf{G}}^P|^2} d\right) \left(1 - \frac{\sum_{\mathbf{G}} |E_{\mathbf{G}}^F|^2 / z_G^F}{\sum_{\mathbf{G}} |E_{\mathbf{G}}^F|^2} d\right) \sim 1 - \left[\frac{\sum_{\mathbf{G}} |E_{\mathbf{G}}^P|^2 / z_G^P}{\sum_{\mathbf{G}} |E_{\mathbf{G}}^P|^2} + \frac{\sum_{\mathbf{G}} |E_{\mathbf{G}}^F|^2 / z_G^F}{\sum_{\mathbf{G}} |E_{\mathbf{G}}^F|^2} \right] d,$$

which is linear in d as well. With larger d , the above integration can be expanded to the next order with another quadratic term, as shown in Fig. S8 (b).

Reviewer Comment:

How does the location (ridge or groove) and density of QDs affect this enhancement?

Our reply:

(a) Location effects:

As figure 2. b and d demonstrate how the relative location of a QD on the PC (ridge or groove) impacts the enhancement factors, the near-field enhancement is slightly stronger in the groove regions. Meanwhile, the ridge regions provide higher Purcell enhancement than the grooves and reach the largest value on the grating edges. In the simulation, we take the average enhancement value among different locations and take the spatial distribution effect into account.

The location effects are now highlighted in the main text, with the following sentence:

(Page 13, line 340): ...Based on the AFM image on the bottom panel, we align the Purcell factor 2D mapping to the topographic structure of the PC grating surface. When the QDs are resonant with the PC fluorescent extraction mode, the ridge regions provide higher Purcell enhancement than the grooves and reach the largest value on the grating edges...

(b) Density effects:

QDs are spatially separated without overlapping, and do not form a continuous layer. Thus, the size uniformity and density of QDs will not affect the enhancement factor. More specifically, the excitation enhancement factor has been defined as the ratio of the power absorbed by a single

QD when it is on PC versus on glass. $\Lambda_{excitation} \equiv \frac{P_{PC}^{abs}}{P_{Glass}^{abs}} \propto \frac{\alpha^P (Q^P)^2}{Q_r^P}$. Although, the size of QD

will affect the absorption cross-section of fluorescent particles, σ_{QD} , this term will be canceled out since it is a common factor in both the numerator and the denominator, and this will not enter into the determination of enhancement factor.

Reviewer Comment:

On page 24 line 496, the authors state that without the target miRNA, the QDs will not be pulled down to the PC surface and will not experience 3000X enhance. What is the EM decay length of PC? If free QDs are present near the PC surface, will the enhancement the same? A control experiment needs to be performed to confirm this.

Our reply:

All QDs present near PC surface will be enhanced and that is why we performed surface blocking and 2-time washing before imaging to minimize non-specific binding events. Based on the counting results of negative control experiments with no target added, the blocking and washing steps have effectively minimized non-specific blinding, as shown in Figure 4 c-f.

The EM decay length of the PC is discussed in SI part 11 and the total enhancement decay is shown in Figure S8.b, where it extends ~60 nm above the PC surface both in the grooves and upon the ridges. This value is now highlighted in the main text, with the following sentence:

(Page 21, line 551): ...Since the PC exclusively enhances QD-tagged probes in close proximity to the surface through evanescent field enhancement and enhanced guided extraction, PCEF enhanced microscopy provides TIRF-like z-sectioning that only enables counting of QD tags within ~60 nm to the PC surface...

Reviewer Comment:

(a) In Fig. 1e (NA/BFP) and Fig.S3, the authors show their experimental setup, in which the sample is illuminated at normal incidence and the resultant PL was measured. (b) This is acceptable only when the luminescence pattern (meaning PL intensity as a function of emission angle) remains unchanged from sample to sample. The PC can exhibit strongly modified luminescence pattern due to diffraction. Therefore, the PL should be measured using an integrating sphere. If this is not possible, the authors should at least provide the PL intensity at various angles and show that the measured photon intensity enhancement has properly taken into account any variations in luminescence pattern.

Our reply:

(a) The laser excitation was at one specific incident angle for pump-mode resonance.

Shown in Fig. S3 (repeated above to assist the reviewer), the laser incident angle is tunable. The subfigure in the bottom right corner (SI Fig. S3) describes how the incident angle θ_{inc} can be precisely tuned by translationally shifting the focus line (Δx displacement) on the BFP through Fourier transform. In order to have enhanced excitation effects, the θ_{inc} is approximately 9.2 degrees rather than at normal incidence to match the coupling condition for laser wavelength (450nm) and excite the pump-mode.

Supplementary Figure S3. Photonic Crystal Enhanced Fluorescence (PCEF) Optical Setup.

Laser: laser diode (450 nm), CL: collimating lens, HWP: half-wave plate, LP: linear polarizer, CYL: cylindrical lens, DM: dramatic Mirror, OBJ: objective Lens, TL: tube lens, M1, M2: Mirrors, EF: emission filter (500 nm long-pass), BFP: back focal plane.

(b) The emission photons are collected in a wide range of solid angles by objective lens.

Our data incorporates measurements of PL intensity at various spatial angles, and we do properly take the “luminescence pattern effect” into account when evaluating the PL enhancement factors. Indeed, when the QD is placed on the PC surface, its luminescence (with a spectral range centered at 605nm) has a spatial distribution pattern caused by the PC dispersion. In fact, this highly directional angular distribution pattern is a major reason for the observed photon collection efficiency enhancement. The objective lens will collect all QD luminescence within its maximal solid angle (limited by its numerical aperture (NA)). In other words, in a microscopy imaging system, we did collect the PL intensity from a wide range of spatial angles by using an objective lens.

A more comprehensive understanding of the PC radiation pattern lies in its isofrequency contours: slices in two-dimensional (2D) momentum space (k_x, k_y) of constant frequency ω ⁸. In order to visualize the QD PL intensity distribution as a function of emission angle in space, we chose to image the back focal plane of the objective lens, because the back focal plane image will provide more information for the readers to not only see the PL intensity distribution pattern (in k_x, k_y spatial domain) but also understand the collection efficiency enhancement owing to the spatial pattern change of QD luminescence.

Figure 3d (repeated below to assist the reader) shows (1) QD luminescence spatial distribution at momentum space as its radiation pattern at the BFP and (2) the QD PL angular distribution

compared to different collection ranges covered by objective lenses.

Figure 3d. BFP images and angular distribution of the emission. Theoretical (red) and experimental measured (blue dots) angular emission intensity for a single QD-605 (center at 604.4nm with FWHM range from 590.4-618.4nm). Insets: Theoretical (red, NA=1) and experimentally measured (blue, NA =0.8, in air) back focal plane images of the single QD. The two vertical dashed lines indicate the maximum collection angle of the objectives (NA=0.5 and NA=0.8).

Reviewer Comment:

...and show that the measured photon intensity enhancement has properly taken into account any variations in luminescence pattern.

Our reply:

We thank the reviewer for pointing out the importance of keeping the same coverage of PL collection range when comparing the photon intensity for evaluating the enhancement factor. When evaluating the enhancement factors, we do properly take the radiation pattern effect into account by using the same NA objective lens that covers the same range of spatial angles instead of a point collector like an optical fiber probe. Meanwhile, the luminescence pattern is determined by the PC designed dispersion band diagram, and irrelevant to QD size or its distribution.

When probing the enhancement factor of extraction rate and quantum efficiency, an oil-emissive objective lens with NA=1.46 (half collection angle ~76 degree) has been used to compare the single QD intensity when it presents on glass or PC (laser incident at 20°, pump-mode off resonance). Since most of the light radiating from the bottom side of glass or PC can be collected in such a large angle coverage, the collection efficiency improvement caused by PC-enabled directional emission (so-called engineered spatial luminescence pattern) is negligible. Moreover, using an objective lens (rather than an integrating sphere) is also a more practical method to examine the enhancements offered by the PC, since normal microscopy imaging system will

never be able to collect the QD luminescence within 4π steradian using an integrating sphere, due to the limited aperture size.

To clarify, we prepared the attached table that summarizes the experimental condition we used to evaluate each enhancement factor. The table is for reviewers' convenience to assess our work.

	Pump-mode resonance (enhance excitation)	Fluorescent mode resonance (enhance extraction rate)	Purcell effect (quantum efficiency enhancement)	Collection efficiency improvement from engineered spatial luminescence pattern
Enhance Excitation Factor	QD on PC laser incident angle: 9.2° Objective lens: 50X, NA = 0.5, air	✓	✓	✓
	QD on PC laser incident angle: 20° Objective lens: 50X, NA = 0.5, air	✗	✓	✓
Enhance Extraction Factor + Enhance Q.E. Factor	QD on PC laser incident angle: 20° Objective lens: 50X, NA = 1.46, oil	✗	✓	✗ (neglectable)
	QD on Glass Objective lens: 50X, NA = 1.46, oil (When QD on glass, we were not able to identify single QD with 50X, NA-0.5 obj.)	✗	✗	✗

Reviewer Comment:

On page 25 line 507, the authors state that if the number of bases of the capture changes from 10 to 7, the capture forms a “hairpin loop” structure and has no ability to pair with the bottom part of the target miRNA. This is surprising because a longer sequence typically tends to fold but the authors postulate otherwise. A short one probably still binds to the target with less affinity. Again, control experiments to support this postulation are needed.

In the miRNA Bridge Assay application, the authors used the oligo pairs (probe-target) design to ensure the specificity. Some information provided is not consistent. Figure 3b shows the schematic of double hybridization, which is not consistent with the sequence description in the supporting information. The information is critical for data interpretation of kinetic results described in Figure 4c-4i. Following the 3' and 5' described in Figure 3b, the mutated base in the target miRNA is located in the region which interacts with the capture. It is puzzling why this induces the kinetic behavior of probe-target interactions. It seems that the interpretation of the kinetic data is based on the mismatch is between the probe and target shown in the Figure S8c. Did these experiments involve washing?

Our reply:

We thank the reviewer for noticing an error in our text describing the oligo sequence, which should have been described as follows: “If the number of bases of the capture sequence increases from 10 to 12, the capture sequence forms a “hairpin loop” structure (shown below) and has no ability to pair with the bottom part of the target miRNA.

The mutated bases in the target miRNA are located in a region that interacts with the probe oligo. All the kinetic behavior experiments did not involve washing step, since washing can cause strong perturbation to transient interaction between the QD probe and the captured target and thus disturb dynamic trajectory analysis.

The following text was modified in the manuscript:

(SI-Page 19, line 316): The oligonucleotides (oligos) used in this study were purchased from Integrated DNA Technologies (IDT DNA Inc, Coralville, Iowa). The probe oligo is functionalized with a biotin at the 3'-end, followed by high-performance liquid chromatography (HPLC) purification. The PC capture probe oligo carries a 5'-primary amine for immobilization to PC surface. Sequences of all the relevant oligos are shown as follows:

PC capture:	5'- NH ₂ -TTTTTTCACGCGAGC-3'
Probe:	5'-CGAACGAACAAATTTT-Biotin-3'
Target miR-375 (PM):	5'-UUUGUUCGUUCGGCUCGCGUGA-3'
4st nucleotide mutation (1MM ₄):	5'-UUU A UUCGUUCGGCUCGCGUGA-3'
6st nucleotide mutation (1MM ₆):	5'-UUUGU C CGUUCGGCUCGCGUGA-3'
8st nucleotide mutation (1MM ₈):	5'-UUUGUUC C UUCGGCUCGCGUGA-3'

(Page 21, line 548): “For example, if the number of bases of the capture sequence increases from 10 to 12, the capture oligos forms a “hairpin loop” structure and has no ability to pair with the bottom part of the target miRNA.

Reviewer comment:

In some experiments, each incubation step is accompanied by gentle washing steps to remove the free targets and unbound QDs was described on page 25 line 521. However, the ability of discriminating a single base difference in a target miRNA in 10 minutes without a washing step was described in the introduction on page 7 line 158. Detailed experimental description is needed to support such claims since washing procedures are generally important in biosensing experiments.

Our reply:

For all experiments, we wash out free targets after incubating with the functionalized PC surface, then a same amount of QD probes is added for tagging the bound targets.

For all counting based experiments, we exam the “end-point value” when the bridge attachment reaction reaches equilibrium. In those experiments, we use washing steps to remove the unbound QDs before image and scanning to minimize the non-specific binding events.

However, when analyzing the dynamic characteristics and comparing the transient interaction between the QD-probe and the PC surface, we did not perform washing after adding the QD-probe, mainly because it will perturb the transient interaction and disturb the dynamic trajectory analyze.

Reviewer comment:

On page 28 line 569, the authors state that three independent double-blind experiments have been carried out. However, no discussion on the statistical analysis of the results.

Our reply:

Three independent double-blind experiments have been carried out during target concentration labeling, assay incubation, and particle counting processes to minimize bias and other anthropic factors. Following by reviewer’s suggestions, we have performed several statistical analyses of those results and added discussion to the main text.

For our dose-response experiments, we first performed one-way ANOVA test to determine whether there are any statistically significant differences between the means of negative control group and 9-testing groups with different target concentrations. The resulted $P < 0.0001$ shows the testing groups have statistically significant differences from the negative control. We then fitting this dose-response curve using log-log linear model with $R^2 = 0.9825$ (SI. Part 15).

For selectivity experiments of single-base variation, we performed four individual t-tests to determine whether two populations are statistically different from each other. Below shown the statistical analysis where **** indicates $P < 0.0001$.

Reviewer comment:

Multiple grammar mistakes need to be corrected during the revision.

Our reply:

All the co-authors have read and carefully edited the revised manuscript to minimize grammatical errors.

Reviewer #3 (Remarks to the Author):

The paper is interesting, providing an ultrasensitive platform for biomedical analysis. It comprises a wealth of data on the analysis of single QD particles for high sensitivity measurements using photonic crystals. The questions that I have are both general, around the structure of the paper, presentation and novelty, as well as specific questions relating to key data.

We thank the reviewer for the complement of our work!

Reviewer comment:

Firstly, there is a general question as to the value of this technology, analytically. With sample preparation taking several hours and measurement then taking an addition 2 hours, how valuable is this to clinical practice. The authors should more clearly articulate a vision as to why this is important to the wider readership of Nature Comm (put another way, what is the clinical decision and care pathway that will be critically affected by whether there is 10aM or 100aM of miRNA in a sample?)

Our reply:

We thank the reviewer for the comment! We have now added the following text and references to the Introduction of the revised manuscript to better explain the motivation and clinical significance of our ultrasensitive miRNA detection platform:

(Page 5, line 137): ... With the prominent rise of liquid biopsy, miRNAs can serve as a promising clinical cancer biomarker, with many studies correlating miRNA concentration to specific health conditions such as a particular cancer type or metastatic state⁹⁻¹². The repeatedly measure of target miRNA concentration from human serum on a daily basis approaches to establish early cancer detection, monitoring of treatments, prognostication and predicting pre-treatment outcomes further emphasizes the need for inexpensive high-performance assays¹³. Our efforts and others have provided evidence that strategically chosen circulating miRNA biomarker concentrations are linked to clinical outcomes. For example, by sequencing RNA contents of plasma exosomes, our team discovered two miRNAs, miR-375 and miR-1290, that strongly associate with clinical outcome in patients with metastatic prostate cancer at the time of developing castration resistance (mCRPC)¹⁴. The detection of miRNA at a very low concentration and with single-base discrimination without involvement of RNA sequencing that needs sophisticated equipment, large sample volumes, and elaborate sample processing is a huge challenge in clinical diagnostic practice.

Unfortunately, the standard protocol of whole blood RNA isolation and purification followed by target identification by quantitative reverse transcriptase PCR (qRT-PCR) is labor-intensive, requires enzymatic amplification, and can suffer from sequence biases¹⁵. In practice, qRT-PCR assays require unique primer and amplification methods for short miRNA target sequences, which make them fail for the analysis of small volumes¹⁶, while sequencing-based approaches (RNA-Seq) require elaborate sample processing, expensive equipment, long wait times, and bioinformatics expertise, all of which limit their use. For qRT-PCR¹⁷, high sensitivity is achieved through enzymatic amplification which requires both conversion to DNA (reverse transcription) and enzymatic amplification to completion for accuracy. Digital droplet approaches have

significantly improved the quantitative analysis of low-volume biospecimens, however similar challenges limit the readout of qRT-PCR assays of miRNA in droplet format, compounded by the low throughput of droplet partitioning, equipment cost, small dynamic range, and complex data analysis steps. Electrochemical sensor with single molecule approaches are capable of ultrasensitive (< 1 pM)¹⁸ and amplification-free miR detection with a simple read out¹⁹⁻²¹ but have restricted range of operating temperature²². However, developing a molecular diagnostic test that is ultrasensitive and highly target specific is necessary to effectively discriminate nucleic acids of similar sequences at low concentrations. Furthermore, a diagnostic assay that does not require enzymatic amplification is desirable for point-of-care use at room temperature. The development of rapid and cost-effective diagnostics is essential for disseminating technologies for clinical applications in broad point of care settings²³. To address this, we present an ultrasensitive sensing strategy/platform for highly specific detection of a cancer-specific miRNA target by providing “digital” resolution of individual molecules with optically enhanced high signal-to-noise ratio.

Reviewer comment:

I would also argue that the introduction is not written to identify novelty. There are a lot of references cited for every part of the system, so the authors are indicating that the only novelty is in the integration. The authors have to clearly show that this is more than simply the sum of the parts, which is not at all clear in the writing in the current form.

Our reply:

We thank the reviewer for identifying this as the Reviewer #1. We have now revised the introduction to better describe the novelty, motivation, and impact of our detection strategy/platform. For more detail response, please refer to our reply for Reviewer #1 on page 1-3 in this document.

Reviewer comment:

On the subject of (2), the claim of a X3000 fold enhancement is relative to a glass slide as reference measurement. What is more interesting to me as a reviewer and indeed to the readership is what the improvement is, relative to other nanostructured photonic systems, especially those using photonic crystals. Again, a detailed critical view is needed.

Our reply:

We thank the reviewer offering this great suggestion. We added the following discussion to the revised manuscript to compare the enhancement factors with for existing fluorescence enhancements using nanostructured photonic systems. We also added a section to review the previous reports that also used photonic crystals for fluorescence enhancements:

(Page 3, line 60): In recent years, a great deal of research has addressed the problem of enhancing the excitation of single fluorescent reporters, efficiently collecting their photon emission, and generating signals that enable them to be observed in the presence of a variety of noise sources. Total Internal Reflection Fluorescence (TIRF) microscopy, for example, can achieve single fluorophore resolution by using expensive high NA oil-immersion objectives and electron-

multiplying charge-coupled (EM-CCD) cameras and provide over a 30× increase in signal-to-noise ratio^{24,25}. Plasmonic nanostructures²⁶⁻³⁰ have been proven successful for localized enhancement of electric field excitation intensity with a fluorescence enhancement factor from ~100× to ~1000×, although many plasmonic structures suffer from high non-radiative decay due to intrinsic losses in the metal, quenching, and low directionality of emitted photons²⁹. Moreover, the resonance wavelength of such nanostructures is fixed by the size, shape, and material of the nano-resonators. Early approaches for exciting fluorescent reporters with dielectric optical microcavities demonstrate modest emission rate enhancements³¹⁻³³. One limitation of dielectric microcavities is the mismatch between high-Q resonances of the cavity and the spectrally wide emission from inhomogeneously broadened fluorescent emitters at room temperature. Recent reports of electromagnetic field enhancement with plasmonic-dielectric hybrid nano-gap and dielectric nanowire-slabs³⁴ addressed these issues and show ~ 1000× enhancement, but with a small number of highly localized “hot spots” that sparsely-occupy only a small fraction of the total surface area. Nevertheless, precise alignment between fluorescent emitters and cavity modes is required to achieve such high enhancement factors through sophisticated nanofabrication. To overcome these issues, our previous research used a microscopy-based approach for fluorescence enhancement from a PC surface over extended surface areas. A 60-fold increase in fluorescence intensity has been reported from a bulk Cy-5 conjugated streptavidin layer on a 1-dimensional PC³⁵. This enhancement can be improved to 360-fold by coupling the PC leaky mode to an underlying Fabry-Perot type cavity through a gold mirror reflector³⁶. A 108x fluorescence enhancement for a layer of QDs has been reported by using leaky-mode assisted fluorescence extraction from a 2-dimensional PCs surface³⁷. More recently, Yan *et al.* demonstrated a multiple heterostructure PC with a super-wide stopband to achieve broadband fluorescence enhancement of over 100-fold³⁸, which required sophisticated layer-by-layer fabrication of self-assembled 2D colloidal crystal monolayers. Three-dimensional PC structures have also been used to enhance fluorescence. Song *et al.* spin-coated a Ru dye layer on 3D opal PCs composed of multilayers of PMMA spheres to achieve ~320-fold luminescence enhancement with dual-stopband configurations³⁹.

Reviewer comment:

The paper would also greatly benefit from re-organisation of the figures. The multiple panels serve only to confuse the reader - there is a strong case for some of these to be moved to ESI (especially in Fig 1., as the panels do not evidence the major point of the paper). I think the authors need to make decisions on key data that corroborates the message(s) in the paper. I believe the paper would be more understandable if Figure 3 was actually Figure 1 and was referenced in the introduction.

Our reply:

We thank the reviewer for such an important suggestion! We have now reorganized the figure sets to provide a better logical flow of information. More specifically, Figure 1 is now a conceptual representation of each of the enhancement effects, and also incorporates the concept of operation for the miRNA assay. Figure 2, in addition to an SEM image of the PC surface with QDs on it, highlights simulation-based analysis that describes each of the enhancement effects, including the PC band diagram that explains the wavelength-angle combinations that lead to formation of resonant modes for excitation and extraction. Figure 3 summarizes experimentally measured data, focused upon single-QD effects of enhanced signal-to-noise, blinking suppression, lifetime and Purcell effect, where we compare the behavior of QDs on the PC to QDs on a glass surface. Figure

4 summarizes the miRNA assay results to show the dose-response plot, selectivity against single-base mismatches, and mean free path observation of QDs being localized by their capture. Other figures (panels) that support the main text data is relegated to the Supplementary Information.

Figure 1. PC enhanced QD emission enables single QD digital counting resolution for single target miRNA detection.

Figure 2. PC-QD system and simulation-based analysis for each enhancement effect.

Figure 3. Experimental data for PC enhanced single-molecule digital sensing.

Reviewer comment:

Most of my specific comments are focused around Figure 4 – which I believe is the key result. As advice, by combining both static and dynamic measurements into a single panel this obscures and confuses – this data should sit within 2 separate figures, associated with the two sections in the text. This is the key experimental evidence and readers must be allowed to understand it.

Our reply:

We have taken the reviewer’s suggestion and are now plotting the static and dynamic measurement data in two separate figures. We kept the Fig.4 (c-f) in main text to summarize the static miRNA assay results. The dynamic measurements for tracking single QD trajectories upon surfaces have been moved to Supplementary Information Fig. S15.

(Page 23, line 581):

Figure 4. Counting result of miR375 bridge assay in human serum. (a) In the bridge activated assay process, QD-tags will be pulled down to the PC surface when target miRNAs bridge the formation of a surface bound complex. (b) Illustration of the line-scanning process that counts the number of PC-attached QDs. (c) Line-scanning Imaging: for PCEF enhanced digital counting in human serum-Target concentration: 10 aM to 1 nM. FOV: 300 μm *300 μm . Scale bar: 20 μm . Data are averages from more than 9 FOVs, and error bars indicate the standard deviation between 3 independent replicas. Statistical significance was tested using one-way ANOVA between negative control group and all testing groups with $P < 0.0001$; Imaging conditions for the assay on the PC: laser power = 1 mW, EM-gain = 40X; Higher laser power (5 mW) and EM-gain (1200X) are needed in order to detect the intensity change at 1-10 pM on glass while avoiding signal saturation at 1 nM; Same integration time (600 ms) and objectives (50X, NA=0.5) were applied in both surface assays. (d) Dose-response curve for various concentrations across a 10^9 -fold concentration range after a 2-hour incubation for digital counting results. (e) Single-base mismatch discrimination test. Line-scanning image panel demonstrates digital resolution of captured QDs for target miR375 perfect match (PM) group, versus three different SNVs at 100 pM. (f) Quantification of perfect match target sequence and the SNV cases at 100 pM concentration. Statistical significance was tested using independent t tests; **** $P < 0.0001$. The error bars represent the standard deviation of three independent assays.

(SI-Page 18, line 299):

Figure S15. Single-base mismatch discrimination using QD-trajectories. Trajectories with 225 time-steps (9s total) for both for (a) perfect match (PM) and (b) single-base mismatch (1MM) groups. PM trajectories show a “confined diffusion” pattern (blue) while 1MM groups appear a “random walk” pattern (green). (c) MSD plot respect to τ , the shade region are average values (green dash line: PM; blue dash line: 1MM) pulse with error bands for five trajectories from each testing group.

Reviewer comment:

Figure 4a has different scaling in inset is also difficult to interpret. Both images at the same scale should at least be provided in Supplementary Information, with the same resolution. Rather than see all 9 concentrations (+ background) better to show less at the same resolution to allow for a critical comparison.

Our reply:

We thank reviewer offer this valuable feedback. We replot all QD on glass assay scanning images with same resolution in Supplementary Information Fig S14.

(SI-Page 18, line 293):

Figure S14. Conventional unenhanced analog ensemble intensity results. Line scanning images for performing the same assay on a glass substrate. Target concentration: 10aM to 1nM. FOV: 300μm *300μm. Scale bar: 20um. Data are averages from more than 9 FOVs, and error bars indicate the standard deviation between 3 independent replicas.

Reviewer comment:

Figure 4b is too difficult to understand with the double axis with different scales. I suspect the reference should also be the background unmodified glass (grey), not the noise (blank +3sd, blue), so that the authors show how much better their device performs relative to an unstructured glass slide. The signal to background would be 3 sd above the grey, not blue, line.

We agree with the reviewer that plotting two dose response curves containing digital counts and analog intensity in same graph could be confusion. Thus, we replot them in two separate figures in the revised manuscript. We believe this is correct to using the background counts from negative control group within the same sensing system/platform (PC-enabled digital counting) for reporting the detection limit. The plot on the right (our comparison for performing the same assay on a glass surface, using analog fluorescent intensity measurements) is referenced in the main body of the manuscript, and now appears in the Supplementary Information (Fig.S14 b).

Reviewer comment:

Fig 4 b, Why did the experiment stop at 10 aM? Given the linear plot over 9 orders of magnitude concentration, and that the signal should still be apparent above the background at 2 orders of magnitude below the current LOD, at 100zM, why did the authors not extend this investigation. If the linear relationship does indeed extend at lower concentrations against the background, this should be shown. Similarly if it does not, it should still be shown so as to give a better understanding of the system.

Our reply:

Following the suggestion, we performed additional experiments to examine the signal at 100 zM and 1 aM of the target miRNA in triplicates in a crude human serum matrix. As shown in the figure below, at such a low concentration the counting results of the detected signal is approximately at the same level as that of the control case, and the two groups cannot be distinguished with statistical significance. Moreover, the clinically relevant concentration of miRNA 375 is at 100 aM-10 pM region⁴⁰, indicating that the sample concentration range tested in our study makes practical sense.

We included a short discussion in the main text correspondingly on this observation:

(Page 25, line 625): ... Further reduction in target concentration (100 zM and 1aM) leads to counting results at the same level of the background signal, as the availability of the surface captured target is now limited by the sample volume. The dilution-error in such low concentration also yields conspicuously high standard deviation.

Reviewer comment:

The same is true at the high concentrations to provide a complete understanding of the technique. Why are measurement above and below the linear range not shown.

Our reply:

The highest clinically relevant concentration of mi-RNA375 (pM level) is already 2-3 orders of magnitude lower than our highest concentration test (1 nM)⁴⁰. Also, the QD probe concentration we used in this system is ~70nM which is limited by the stock concentration of commercial QD products. In order to make sure the QD probe is sufficient for highest target concentration, we chose the 1 nM to be the greatest concentration tested.

Reviewer comment:

Figure 4c-d are unclear (the positioning of the particles looks exactly the same and the cross covers a lot more than a single base). The Figure certainly does not give credit to the 22 different possibilities that the authors have analysed.

Our reply:

We revised our figure (related to the single nucleotide variation, SNV) to better display the locations of each of the single nucleotide mismatch on the target miRNA chain.

Reviewer comment:

As a final comments, some thought about significant figures and errors – e.g 1.708ns (+/-0.3nS). Does this measurement really have a precision around 1 ps or is this algorithmic from the fitting function used in Fig 2(g), and what does that value mean when the error is 2 orders of magnitude higher than the measurement. Do the authors mean 1.7 ns (+/-0.3nS) which is less convincing, or is this a typographic error e.g. 1.708ns (+/-0.3pS).

Our reply:

We thank the reviewer for helping us correct the proper use in significant figures. The photon counting system we use have about 0.022ns (22ps) time-resolution, the reported average value of “1.708ns” was the algorithmic value from the fitting function with a variation range of +/- 0.31nS. We have now revised the lifetime discription as flowing:

(Page 18, line 473): Similar to the numerical simulation, our Time-Resolved Photoluminescence (TRPL) measurement shows an experimental Purcell factor of ~3.11x for ensemble QDs on the PC surface compared to glass. Shown as Fig. 3c, the averaged decay time of $\tau_{\text{glass}} = 5.32\text{ns}$ ($\pm 0.24\text{ns}$) while the $\tau_{\text{pc}} = 1.71\text{ns}$ ($\pm 0.31\text{ns}$).

Reviewer comment:

Similarly in Figure 4(i), the reader should have a view of variability (how do the 5 trajectory measurements compare)? Another example of important information almost lost by the compression of 9 figures into one panel, and by merging the static and dynamic data in the same figure.

Our reply:

We thank reviewer offer this valuable feedback. We plot out the MSD measurement of 5 trajectories for both perfect-match group (PM) and mismatch group (MM) and move the dynamic measurement of tracking single QD trajectories to SI. Fig S15.

Figure S15. Single-base mismatch discrimination using QD-trajectories. ... (c) MSD plot respect to τ , the shade region are average values (green dash line: PM; blue dash line: IMM) pulse with error bands for five trajectories from each testing group.

References:

- 1 Ganesh, N. *et al.* Enhanced fluorescence emission from quantum dots on a photonic crystal surface. *Nature Nanotechnology* **2**, 515-520, doi:10.1038/nnano.2007.216 (2007).
- 2 Chen, X. *et al.* Characterization of microRNAs in serum: a novel class of biomarkers for diagnosis of cancer and other diseases. *Cell Research* **18**, 997-1006, doi:10.1038/cr.2008.282 (2008).
- 3 Cai, S. *et al.* Single-molecule amplification-free multiplexed detection of circulating microRNA cancer biomarkers from serum. *Nature Communications* **12**, 3515, doi:10.1038/s41467-021-23497-y (2021).
- 4 Liu, G. *et al.* CircFAT1 sponges miR-375 to promote the expression of Yes-associated protein 1 in osteosarcoma cells. *Molecular Cancer* **17**, 170, doi:10.1186/s12943-018-0917-7 (2018).
- 5 Che, C. *et al.* Activate capture and digital counting (AC + DC) assay for protein biomarker detection integrated with a self-powered microfluidic cartridge. *Lab Chip* **19**, 3943-3953, doi:10.1039/c9lc00728h (2019).

- 6 Findlay, J. W. & Dillard, R. F. Appropriate calibration curve fitting in ligand binding assays. *AAPS J* **9**, E260-267, doi:10.1208/aapsj0902029 (2007).
- 7 Squires, T. M., Messinger, R. J. & Manalis, S. R. Making it stick: convection, reaction and diffusion in surface-based biosensors. *Nature Biotechnology* **26**, 417-426, doi:10.1038/nbt1388 (2008).
- 8 Regan, E. C. *et al.* Direct imaging of isofrequency contours in photonic structures. *Science Advances* **2**, e1601591, doi:doi:10.1126/sciadv.1601591 (2016).
- 9 Mitchell, P. S. *et al.* Circulating microRNAs as stable blood-based markers for cancer detection. *P Natl Acad Sci USA* **105**, 10513-10518, doi:10.1073/pnas.0804549105 (2008).
- 10 Bettegowda, C. *et al.* Detection of circulating tumor DNA in early- and late-stage human malignancies. *Sci Transl Med* **6**, 224ra224, doi:10.1126/scitranslmed.3007094 (2014).
- 11 Ortholan, C. *et al.* MicroRNAs and lung cancer: New oncogenes and tumor suppressors, new prognostic factors and potential therapeutic targets. *Curr Med Chem* **16**, 1047-1061 (2009).
- 12 Rosell, R., Wei, J. & Taron, M. Circulating MicroRNA Signatures of Tumor-Derived Exosomes for Early Diagnosis of Non-Small-Cell Lung Cancer. *Clin Lung Cancer* **10**, 8-9, doi:10.3816/CLC.2009.n.001 (2009).
- 13 Donaldson, J. & Park, B. H. Circulating Tumor DNA: Measurement and Clinical Utility. *Annu Rev Med* **69**, 223-234, doi:10.1146/annurev-med-041316-085721 (2018).
- 14 Huang, X. *et al.* Exosomal miR-1290 and miR-375 as Prognostic Markers in Castration-resistant Prostate Cancer. *Eur Urol* **67**, 33-41, doi:10.1016/j.eururo.2014.07.035 (2015).
- 15 El-Khoury, V., Pierson, S., Kaoma, T., Bernardin, F. & Berchem, G. Assessing cellular and circulating miRNA recovery: the impact of the RNA isolation method and the quantity of input material. *Sci. Rep.* **6**, 19529, doi:10.1038/srep19529 (2016).
- 16 Hunt, E. A., Broyles, D., Head, T. & Deo, S. K. MicroRNA Detection: Current Technology and Research Strategies. *Annu Rev Anal Chem (Palo Alto Calif)* **8**, 217-237, doi:10.1146/annurev-anchem-071114-040343 (2015).
- 17 Hunt, E. A., Broyles, D., Head, T. & Deo, S. K. MicroRNA Detection: Current Technology and Research Strategies. *Annual review of analytical chemistry (Palo Alto, Calif.)* **8**, 217-237, doi:10.1146/annurev-anchem-071114-040343 (2015).
- 18 Wu, Y. F., Tilley, R. D. & Gooding, J. J. Challenges and Solutions in Developing Ultrasensitive Biosensors. *J. Am. Chem. Soc.* **141**, 1162-1170, doi:10.1021/jacs.8b09397 (2019).
- 19 Labib, M., Sargent, E. H. & Kelley, S. O. Electrochemical Methods for the Analysis of Clinically Relevant Biomolecules. *Chem Rev* **116**, 9001-9090, doi:10.1021/acs.chemrev.6b00220 (2016).
- 20 Tavallaie, R. *et al.* Nucleic acid hybridization on an electrically reconfigurable network of gold-coated magnetic nanoparticles enables microRNA detection in blood. *Nat. Nanotechnol.* **13**, 1066-1071, doi:10.1038/s41565-018-0232-x (2018).
- 21 Johnson-Buck, A. *et al.* Kinetic fingerprinting to identify and count single nucleic acids. *Nat. Biotechnol.* **33**, 730-732, doi:10.1038/nbt.3246 (2015).
- 22 Ronkainen, N. J., Halsall, H. B. & Heineman, W. R. Electrochemical biosensors. *Chem Soc Rev* **39**, 1747-1763, doi:10.1039/b714449k (2010).
- 23 Vashist, S. K., Lippa, P. B., Yeo, L. Y., Ozcan, A. & Luong, J. H. T. Emerging Technologies for Next-Generation Point-of-Care Testing. *Trends Biotechnol* **33**, 692-705, doi:10.1016/j.tibtech.2015.09.001 (2015).

- 24 Vu, T. Q., Lam, W. Y., Hatch, E. W. & Lidke, D. S. Quantum dots for quantitative imaging: from single molecules to tissue. *Cell and Tissue Research* **360**, 71-86, doi:10.1007/s00441-014-2087-2 (2015).
- 25 Le, P. *et al.* Counting growth factors in single cells with infrared quantum dots to measure discrete stimulation distributions. *Nature Communications* **10**, 909, doi:10.1038/s41467-019-08754-5 (2019).
- 26 Kinkhabwala, A. *et al.* Large single-molecule fluorescence enhancements produced by a bowtie nanoantenna. *Nature Photonics* **3**, 654-657, doi:10.1038/nphoton.2009.187 (2009).
- 27 Punj, D. *et al.* Self-Assembled Nanoparticle Dimer Antennas for Plasmonic-Enhanced Single-Molecule Fluorescence Detection at Micromolar Concentrations. *ACS Photonics* **2**, 1099-1107, doi:10.1021/acsp Photonics.5b00152 (2015).
- 28 Zhang, T., Gao, N., Li, S., Lang, M. J. & Xu, Q.-H. Single-Particle Spectroscopic Study on Fluorescence Enhancement by Plasmon Coupled Gold Nanorod Dimers Assembled on DNA Origami. *The Journal of Physical Chemistry Letters* **6**, 2043-2049, doi:10.1021/acs.jpcclett.5b00747 (2015).
- 29 Akselrod, G. M. *et al.* Probing the mechanisms of large Purcell enhancement in plasmonic nanoantennas. *Nature Photonics* **8**, 835-840, doi:10.1038/nphoton.2014.228 (2014).
- 30 Luan, J. *et al.* Ultrabright fluorescent nanoscale labels for the femtomolar detection of analytes with standard bioassays. *Nat Biomed Eng* **4**, 518-530, doi:10.1038/s41551-020-0547-4 (2020).
- 31 Liu, J. N., Huang, Q., Liu, K. K., Singamaneni, S. & Cunningham, B. T. Nanoantenna-Microcavity Hybrids with Highly Cooperative Plasmonic-Photonic Coupling. *Nano Lett* **17**, 7569-7577, doi:10.1021/acs.nanolett.7b03519 (2017).
- 32 Joannopoulos, J. D. *Photonic crystals : molding the flow of light*. 2nd edn, (Princeton University Press, 2008).
- 33 Fan, S., Suh, W. & Joannopoulos, J. D. Temporal coupled-mode theory for the Fano resonance in optical resonators. *J Opt Soc Am A Opt Image Sci Vis* **20**, 569-572, doi:10.1364/josaa.20.000569 (2003).
- 34 Kolchin, P. *et al.* High Purcell Factor Due To Coupling of a Single Emitter to a Dielectric Slot Waveguide. *Nano Letters* **15**, 464-468, doi:10.1021/nl5037808 (2015).
- 35 Ganesh, N. *et al.* Leaky-mode assisted fluorescence extraction: Application to fluorescence enhancement biosensors. *Opt. Express* **16**, 21626-21640, doi:10.1364/OE.16.021626 (2009).
- 36 Pokhriyal, A., Lu, M., Chaudhery, V., George, S. & Cunningham, B. T. in *CLEO: 2013*. 1-2.
- 37 Ganesh, N. *et al.* Enhanced fluorescence emission from quantum dots on a photonic crystal surface. *Nat Nanotechnol* **2**, 515-520, doi:10.1038/nnano.2007.216 (2007).
- 38 Zhang, L., Wang, J., Tao, S., Geng, C. & Yan, Q. Universal Fluorescence Enhancement Substrate Based on Multiple Heterostructure Photonic Crystal with Super-Wide Stopband and Highly Sensitive Cr(VI) Detecting Performance. *Advanced Optical Materials* **6**, 1701344, doi:10.1002/adom.201701344 (2018).
- 39 Eftekhari, E. *et al.* Anomalous Fluorescence Enhancement from Double Heterostructure 3D Colloidal Photonic Crystals—A Multifunctional Fluorescence-Based Sensor Platform. *Scientific Reports* **5**, 14439, doi:10.1038/srep14439 (2015).
- 40 Huang, X. *et al.* Exosomal miR-1290 and miR-375 as prognostic markers in castration-resistant prostate cancer. *Eur Urol* **67**, 33-41, doi:10.1016/j.eururo.2014.07.035 (2015)

REVIEWER COMMENTS

Reviewer #1 (Remarks to the Author):

The authors have thoroughly addressed all of my comments except the question regarding assay linearity (Fig. 4b). The fact that the data look linear on a log-log plot does NOT AT ALL mean that the functional dependence is linear. It is quite obviously not. When the target concentration is increased by a factor 100 million, the number of QD counts only increases by a factor of 10. Why is that? Looking at the lower concentrations and the error bars of the assay, this seems to imply that a 10x increase in target concentration results only in a few (~3-5) more counts. How can counts be a reliable measure of concentration in this case? The discussion of the reasons and implications of this behavior should be added to the manuscript.

Reviewer #2 (Remarks to the Author):

The authors significantly improved the paper's quality by addressing the reviewers' comments. The article is recommended for publication in its current form.

Reviewer #3 (Remarks to the Author):

The paper is much improved and in fact the authors address all three referee's comments well. I am very happy to accept it.

As a small note - some of the English/syntax needs some care/editing where new text has been included. For example the sentences below could be improved:

The repeatedly measure of target miRNA concentration from human serum on a daily basis approaches to establish early cancer detection, monitoring of treatments, prognostication and predicting pre-treatment outcomes further emphasizes the need for inexpensive high-performance assays

similarly

Electrochemical sensor with single molecule approaches are capable of ultrasensitive

should read sensors

Finally, if the authors need to cite a miRNA isothermal assay in their discussion (they mention PCR but not isothermal methods), please feel happy to include "Programmable design of isothermal nucleic acid diagnostic assays through abstraction-based models" as a new Ref 15 (Nature Communications volume 13, Article number: 1635 (2022)).

The authors are grateful for the additional comments offered by Reviewer #1 and Reviewer #3. We note that Reviewer #2 indicated their satisfaction with the manuscript. We have made further changes to the manuscript to clarify these issues for the readership. In the following narrative, reviewer comments are shown in black text, and our response is shown in blue text. We indicate where changes have been made to the manuscript.

Reviewer #1

Reviewer Comment:

The authors have thoroughly addressed all of my comments except the question regarding assay linearity (Fig. 4b). The fact that the data look linear on a log-log plot does NOT AT ALL mean that the functional dependence is linear. It is quite obviously not. When the target concentration is increased by a factor 100 million, the number of QD counts only increases by a factor of 10. Why is that? How can counts be a reliable measure of concentration in this case?

Our reply:

We agree with the reviewer that our assay response is not strictly linear. Thus, we have removed the phrases “linear dose-response” at line 38, line 601, line 565, and line 712 in our updated manuscript.

We believe that this concern stems from the manner in which we reported QD counts in Figure 4b. Owing to the limited space and figure resolution, we only show a small region ($300\ \mu\text{m}^2$) of the entire sensor surface area ($\sim 2\ \text{mm}^2$). As a result, the plotted value (on the y -axis, labeled “QD Digital Counts”) only represents the QD counts gathered over this small region (which we call “sub-Fields of View (sub-FOV)”) of the overall biosensor surface. Thus, the actual total number of QDs “counted” on the sensor surface is actually $\sim 20\text{X}$ greater than the numerical value shown in Figure 4b. Because the total scanned biosensor surface area is not identical for each data point, it is necessary to apply some type of normalization. Therefore, in the revised manuscript, we changed the y -axis of Figure 4b to “Digitally Counted QD Surface Density” in units of QD/mm^2 .

This question raises a related point that caused us to more carefully consider the issue of counting efficiency of miRNA molecules in the assay. Indeed, when the miRNA concentration increases 10-fold, we do not observe a 10-fold increase in QD counts. Surface-based biosensors have inherent limitations from diffusion, convection, mass transport, surface functionalization density, binding kinetics, steric hindrance, and surface saturation¹⁻⁴, where only a fraction of the total target molecules in solution are captured on the sensing surface and subsequently labeled by fluorescent probes. In our experiments, the total scanned surface area (~9 sub-FOV) of the PC biosensor represents only 45% the PC area in the bottom of our fluid compartment, and thus ~55% of the potentially available QD-tagged miRNA are not counted. Assuming a uniform distribution of QDs across the available biosensor surface (scanned and unscanned) enables us to estimate the overall capture/counting efficiency by dividing the (estimated) total captured QD-tagged miRNA by the total number of target miRNA molecules in the well. This estimate reveals an average 14.2% miRNA counting efficiency, which represents a very good value for surface-based nucleic acid capture efficiency⁵. Looking at the lower concentrations of the assay, for example, we observe that an increase from 10 aM to 100 aM results in ~250 additional QD counts when scanning the entire sensor surface, where the number of miRNA molecules in the well should increase by ~1200.

Considering this question from a statistical perspective regarding differentiation of “detected” concentrations from the negative control, we performed one-way ANOVA analysis to determine whether there are any statistically significant differences between the means of the negative control group and the

9-testing groups with different miRNA concentrations. The resulted $P < 0.0001$ shows the testing groups have statistically significant differences from the negative control and the established calibration curves (Fig. 4b) demonstrate no significant overlap between adjacent concentrations. We then fit this dose-response curve using a log-log linear model with $R^2 = 0.9825$ (SI. Part 15).

Reviewer Comment:

The discussion of the reasons and implications of this behavior should be added to the manuscript.

Our reply:

The following discussion has been added to the discussion section of our revised manuscript.

(Line 685): “We observe a “linear” dose-response plot when our data is plotted on a log-log scale (Fig. 4b). The observed behavior is the result of several factors that include: (a) the limited diffusion¹ of the miRNA to the biosensor surface, where they can be captured, especially at lower concentrations, and (b) the steric hindrance and surface saturation, where the fraction of available binding sites on the PC surface is reduced at high concentrations, making it more difficult for newly-arriving miRNA to bind^{3,6}. Importantly, we note that the dose-response plot (Fig. 4b) demonstrates no significant overlap between adjacent concentrations based on the standard deviation values for three independent measurements at each concentration. Thus, the digital resolution detection approach shown in our work does not require Poisson correction, as needed for approaches such as ddPCR and Quanterix Simoa™, where multiple target molecules confined in the same nanodroplet volume are amplified together while still yielding only one “positive” event.”

Reviewer #3:

Reviewer Comment:

The paper is much improved and in fact the authors address all three referee's comments well. I am very happy to accept it.

As a small note - some of the English/syntax needs some care/editing where new text has been included. For example the sentences below could be improved:

“The repeatedly measure of target miRNA concentration from human serum on a daily basis approaches to establish early cancer detection, monitoring of treatments, prognostication and predicting pre-treatment outcomes further emphasizes the need for inexpensive high-performance assays”

Similarly,

“Electrochemical sensor with single molecule approaches are capable of ultrasensitive”
should read sensors

Our reply:

We thank the reviewer for pointing out grammatical errors. We have carefully reviewed and edited the final version for correct grammar.

Reviewer Comment:

Finally, if the authors need to cite a miRNA isothermal assay in their discussion (they mention PCR but not isothermal methods), please feel happy to include “Programmable design of isothermal nucleic acid diagnostic assays through abstraction-based models” as a new Ref 15 (Nature Communications volume 13, Article number: 1635 (2022)).

Our reply:

We thank the reviewer for this great suggestion. We have now incorporated this reference⁷ in main text as Ref 33.

- 1 Squires, T. M., Messinger, R. J. & Manalis, S. R. Making it stick: convection, reaction and diffusion in surface-based biosensors. *Nature Biotechnology* **26**, 417-426, doi:10.1038/nbt1388 (2008).
- 2 Akkilic, N., Geschwindner, S. & Höök, F. Single-molecule biosensors: Recent advances and applications. *Biosensors and Bioelectronics* **151**, 111944, doi:<https://doi.org/10.1016/j.bios.2019.111944> (2020).
- 3 Cretich, M., Daaboul, G. G., Sola, L., Ünlü, M. S. & Chiari, M. Digital detection of biomarkers assisted by nanoparticles: application to diagnostics. *Trends Biotechnol*

- 33**, 343-351, doi:10.1016/j.tibtech.2015.03.002 (2015).
- 4 Schuck, P. & Zhao, H. in *Surface Plasmon Resonance: Methods and Protocols* (eds Nico J. Mol & Marcel J. E. Fischer) 15-54 (Humana Press, 2010).
- 5 Dandy, D. S., Wu, P. & Grainger, D. W. Array feature size influences nucleic acid surface capture in DNA microarrays. *Proceedings of the National Academy of Sciences* **104**, 8223-8228, doi:doi:10.1073/pnas.0606054104 (2007).
- 6 F, E. K. *et al.* Attomolar sensitivity microRNA detection using real-time digital microarrays. *ChemRxiv. Cambridge: Cambridge Open Engage; 2022; This content is a preprint and has not been peer-reviewed.* (2022).
- 7 Xu, G. *et al.* Programmable design of isothermal nucleic acid diagnostic assays through abstraction-based models. *Nature Communications* **13**, 1635, doi:10.1038/s41467-022-29101-1 (2022).

REVIEWERS' COMMENTS

Reviewer #1 (Remarks to the Author):

The authors have addressed all concerns. The article is recommended for publication in its current form.

REVIEWERS' COMMENTS

Reviewer #1 (Remarks to the Author):

The authors have addressed all concerns. The article is recommended for publication in its current form.

Reponses: We thanks the reviewer for reviewing our manuscript.